# Organic electrochemical transistors as on-site signal amplifiers for electrochemical aptamer-based sensing

Xudong Ji [1,2], Xuanyi Lin[3,4,5] & Jonathan Rivnay [1,2]✉

Electrochemical aptamer-based sensors are typically deployed as individual, passive, surface-functionalized electrodes, but they exhibit limited sensitivity especially when the area of the electrode is reduced for miniaturization purposes. We demonstrate that organic electrochemical transistors (electrolyte gated transistors with volumetric gating) can serve as on-site amplifiers to improve the sensitivity of electrochemical aptamer-based sensors. By mono-lithically integrating an Au working/sensing electrode, on-chip Ag/AgCl reference electrode, and Poly(3,4-ethylenedioxythiophene)-poly(-styrenesulfonate) counter electrode − also serving as the channel of an organic electrochemical transistor− we can simultaneously perform testing of organic electrochemical transistors and traditional electroanalytical measurement on electrochemical aptamer-based sensors including cyclic voltammetry and square-wave voltammetry. This device can directly amplify the current from the electrochemical aptamer-based sensor via the in-plane current modulation in the counter electrode/transistor channel. The integrated sensor can sense transforming growth factor beta 1 with 3 to 4 orders of magnitude enhancement in sensitivity compared to that in an electrochemical aptamer-based sensor (292 µA/dec vs. 85 nA/dec). This approach is believed to be universal, and can be applied to a wide range of tethered electrochemical reporter-based sensors to enhance sensitivity, aiding in sensor miniaturization and easing the burden on backend signal processing.

Electrochemical aptamer-based (E-AB) sensors have been widely used in the last two decades to detect a large range of targets from ions[1] and small molecules[2,3] to nucleic acids[4], proteins[5], and whole cells[6]. In addition to the applicability towards various sensing targets, the potential for real-time sensing[7] and in-vivo implementation[2] have motivated E-AB sensor development. Advantages in E-AB sensors are largely due to the merits of aptamers, including their ease of chemical synthesis, strong and tunable binding of specific analytes, wide applicability of different targets, good thermal/environment stability,

fast-production, and low-cost[8,9]. Electrochemical detection offers sensitive readout of binding-induced conformation changes, often transduced via changes in electron transfer between an electrode and a redox reporter bound to the aptamer. E-AB sensors based on a modified electrode, either thin film electrode or bulk wire electrode, have been used as a standard structure in the research community[10]. The aptamer is commonly modified with a thiol group on one end to bond with the electrode, and redox reporter on the other end. Signals are usually transduced using established electrochemical

[1]Department of Biomedical Engineering, Northwestern University, Evanston, IL 60208, USA. [2]Simpson Querrey Institute, Northwestern University, Chicago, IL 60611, USA. [3]Center for Sleep and Circadian Biology, Northwestern University, Evanston, IL 60208, USA. [4]Department of Neurobiology, Northwestern University, Evanston, IL 60208, USA. [5]Department of Psychology, The University of Hong Kong, Pokfulam Road, Hong Kong SAR, China. ✉e-mail: jrivnay@northwestern.edu

interrogation methods like chronoamperometry (CA)[11], cyclic voltammetry (CV)[12], square wave voltammetry (SWV)[13] and electrochemical impedance spectroscopy (EIS)[14]. Although the E-AB sensors have been successfully utilized both in vitro and in vivo[2,11], the current sensitivity is limited by the surface area of the electrode which determines the number of aptamers which generate a signal upon target binding. Thus, there is a trade-off between high sensitivity and device miniaturization.

One strategy to increase the sensitivity of electrode-based E-AB sensors is to create electrodes with high surface area through electrochemical alloying/dealloying[15], surface wrinkling[16] or electrochemical nanostructuring[17]. However, the upper limit for the enhancement of the surface area and the sensitivity can only be improved dozens of times, at most[15]. Another strategy to enhance sensitivity is through amplification, typically by implementing a transistor[18,19]. Among different types of transistors, organic electrochemical transistors (OECTs) have gained particular attention[20,21]. An OECT is a three terminal device composed of a gate, drain, and source terminal, where a mixed ionic-electronic conducting material forms the channel between source and drain[22]. The channel's conductivity can be altered by the ion injection/extraction controlled through the gate bias[23]. Due to its ion-to-electron converting property, high transconductance, and biocompatibility, OECTs have proven attractive as biophysical[24–26] and biochemical[27–31] sensors for a large variety of targets with high sensitivity and on-site amplification. Unlike frequently used biorecognition elements such as ion-selective membranes, enzymes, and antibodies, aptamers are rarely used in OECTs and very few studies have reported integrating OECTs with E-AB sensors[32–34]. A conventional approach to OECT-based transduction of biochemical sensing is to modify the gate electrode of the OECT (functioning as a working electrode), which we denoted as conv-OECT in this work. This conv-OECT has previously been used in OECT-based E-AB sensors, where the Au gate is functionalized by the redox-reporter modified aptamer and the sensor output is considered the shift of the transfer curve of OECT. However, in this scenario, it is difficult for the OECT to capture the modulation of electron transfer kinetics of the redox reporter, which is altered by the structural modulation of the aptamer upon target binding. While operational, many aptamer-based OECT sensors' sensing mechanisms are likely due to small changes in impedance (most likely capacitive) in the ionic circuit between gate and channel after target binding, which results in the shift of their transfer curve. As such, the typical sensing mechanism in E-AB sensors is not harnessed in previous OECT devices, limiting their generalizability. As a result, a redesigned device concept, architecture, and testing scheme is needed to integrate and characterize the OECT-based E-AB sensors. Such a device should fully utilize established sensing mechanisms while taking advantage of the on-site amplification properties of OECTs.

Herein, we develop a referenced-OECT, or ref-OECT-based E-AB sensor by monolithically integrating aptamer-modified Au working/sensing electrode, on-chip Ag/AgCl reference electrode, and Poly(3,4-ethylenedioxythiophene)-poly(styrenesulfonate) (PEDOT:PSS) counter electrode. This device retains the features of both OECT and E-AB sensor, guaranteeing the functionality of both. The operation of the E-AB sensor is based on a typical 3-electrode setup and ensures the applicability of established electroanalytical techniques like cyclic voltammetry (CV) and square-wave voltammetry (SWV), retaining the original sensing mechanism. The conductivity changes of the PEDOT:PSS counter electrode caused by the doping/de-doping process from the ionic current during the operation of the E-AB sensor can be monitored with two additional contact leads, which provide the output of the OECT device. In this way, direct amplification of the current in the working electrode (gate, E-AB sensor) to the in-plane current modulation in the counter electrode (OECT channel) can be achieved. As a proof of concept, the ref-OECT-based E-AB sensor is used to sense transforming growth factor beta 1 (TGF-$\beta_1$), which is one of the most important biomarkers during wound healing process[35], with 3-4 orders of magnitude enhancement in sensitivity (290 μA/dec for CV-ref-OECT, 292 μA/dec for SWV-ref-OECT) compared to the bare E-AB sensor (24 nA /dec for CV, 85 nA/dec for SWV), with similar detection limit (~1 ng/mL). At the same time, ref-OECT-based E-AB sensor also shows enhancement in sensitivity (2.90 mS/dec vs. 0.51 mS/dec) compared to conv-OECT-based E-AB sensor. This approach is believed to be universal since it can be applied to a wide range of tethered redox-reporter-based electrochemical sensors with various electrochemical interrogation methods to enhance sensitivity and improve device form factor and integration.

## Results

### Design concept of ref-OECT-based E-AB sensor

The schematic of the ref-OECT-based E-AB sensor and the testing scheme are shown in Fig. 1a and b. The device is composed of three aptamer-modified Au electrodes, one Ag/AgCl electrode and one PEDOT:PSS electrode, which were monolithically integrated using multiple photolithography, vapor phase deposition and etching processes (Supplementary Fig. 1). From the point of view of E-AB sensor, the Au, Ag/AgCl and PEDOT:PSS electrodes are regarded as the working, reference, and counter electrodes, respectively. In this design, established and accepted electrochemical measurements like CV and SWV can be conducted in the E-AB sensor. From the perspective of the OECT, the Au and PEDOT:PSS can be considered as the gate and channel respectively, Ag/AgCl serves as a reference point to set the precise voltage drop at gate/electrolyte interface. The interdigitated drain and source electrodes are patterned underneath the PEDOT:PSS channel which defines a large $W/L$ (54) to boost the channel current modulation, while another Au electrode beside the drain and source electrodes serves as the connection lead for the counter electrode, as shown in the enlarged view in Fig. 1a. Supplementary Fig. 2 shows the microscope image of the ref-OECT-based E-AB sensor where the Au sensing gates and the on-chip Ag/AgCl reference electrode are both 2 mm × 2 mm. The PEDOT:PSS counter electrode is 240 μm × 240 μm with a channel width of 1080 μm and a channel length of 20 μm. During the operation of the ref-OECT-based E-AB sensor, the traditional electrochemical measurement was conducted in the E-AB sensor with the above-mentioned 3-electrode setup, while the change of in-plane conductivity in PEDOT:PSS is monitored at the same time by the drain and source electrodes and regarded as the output of the OECT device. Since the characterization of the working electrode is performed in a standard 3-electrode setup, the electrochemistry that happens on the working electrode can be retained with respect to the analysis of an electrode-based E-AB sensor despite the integration with an OECT. Considering the circuit from the modified Au working electrode to PEDOT:PSS counter electrode, the current and total charge (time integral of current) passing through the working electrode is equal to that in counter electrode. In this regard, the PEDOT:PSS counter electrode serves as a capacitively coupled electrode whose conductivity can be modulated by the doping/de-doping of the channel active materials resulting from ion injection/extraction, which is driven by the current from working electrode. Since the change of the conductivity in the PEDOT:PSS counter electrode leads to variation of channel current in the OECT, we can directly relate the output of the OECT to the current in working electrode in the E-AB sensor (also considered as gate current $I_G$ from the perspective of the OECT).

To demonstrate the proposed sensing mechanism of the ref-OECT-based E-AB sensor, we take the sensing of TGF-$\beta_1$ as an example as shown in Fig. 1c. E-AB sensors for TGF-$\beta_1$ have been demonstrated previously as a "signal-off" type of sensor[36,37]. In brief, in the absence of TGF-$\beta_1$, the aptamer is in a conformation where the methylene blue (MB) redox reporter is close to the Au surface, which results in a high current in working electrode (high $I_G$) during CV or SWV measurement. The high

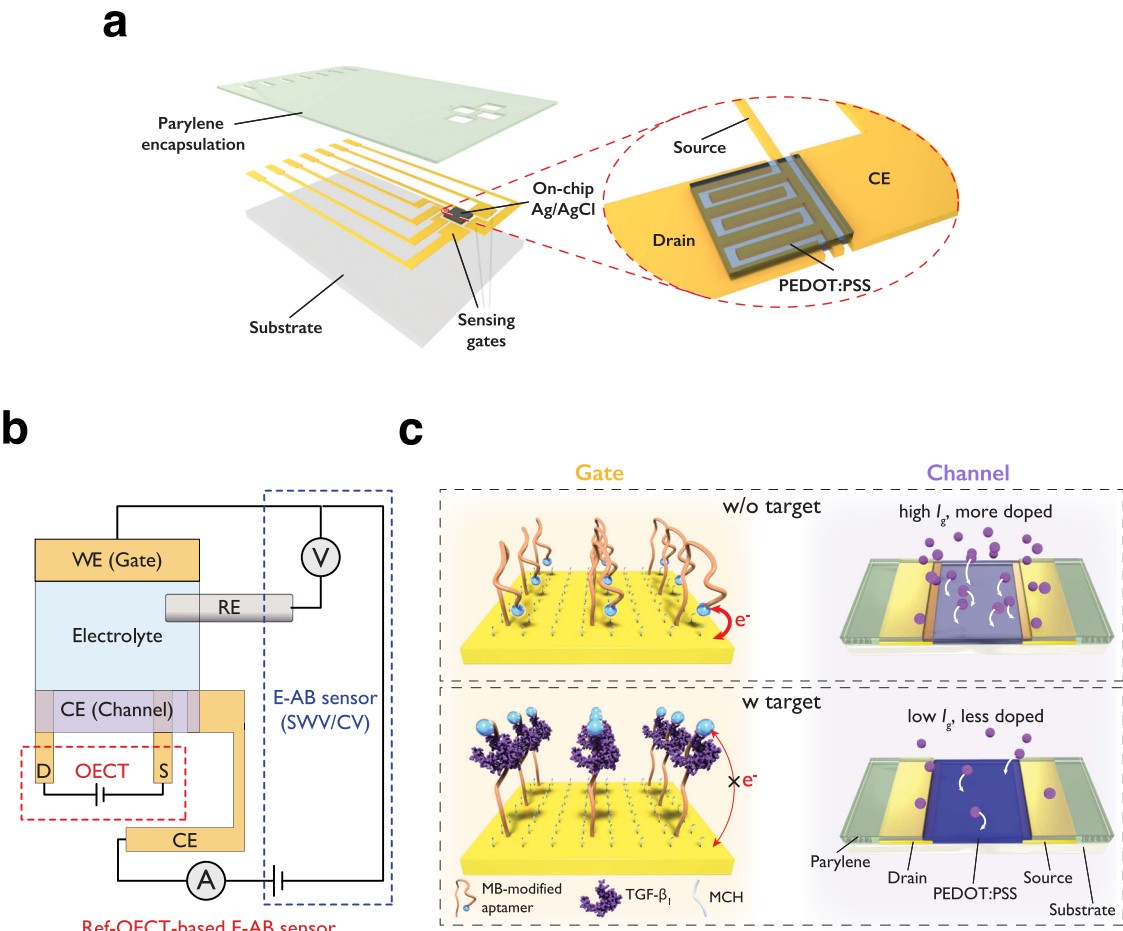

**Fig. 1 | Design concept of ref-OECT-based E-AB sensor. a** Schematic image of the ref-OECT-based E-AB sensor. **b** Testing scheme of the ref-OECT-based E-AB sensor. Output of channel current in OECT can be monitored during the operation of E-AB sensor in the 3-electrode setup. **c** Sensing mechanism of the ref-OECT-based E-AB sensor for TGF-$\beta_1$. Without the existence of TGF-$\beta_1$, the methylene blue (MB) redox reporter is closer to the gate electrode surface, which results in a high gate current ($I_G$) as well as a larger channel current modulation ($I_{DS}$). In the presence of TGF-$\beta_1$, a conformational change occurs in the aptamer, and the MB redox reporter moves further from the gate electrode surface, which results in low gate current and smaller channel current modulation.

$I_G$ will lead to an elevated ion injection into the PEDOT:PSS counter electrode, which extensively alters the doping level of the PEDOT:PSS, inducing a larger channel current modulation recorded by drain/source electrodes. On the other hand, in the presence of TGF-$\beta_1$, the binding between the aptamer and the TGF-$\beta_1$ causes the conformational change of the aptamer and thus moves the redox-active MB away from the electrode surface, which results in a lower current in the working electrode (low $I_G$). This low $I_G$ will lead to less ion injection into PEDOT:PSS and hence a smaller channel current modulation. In this case, the degree of channel current modulation can be regarded as an indicator of the TGF-$\beta_1$ concentration in our ref-OECT-based E-AB sensor.

## Characterization of individual components in ref-OECT-based E-AB sensor

To support our proposed operation and sensing mechanism of the ref-OECT-based E-AB sensor, we first characterize the aptamer functionalization on the Au electrode, which is a critical step for sensing selectivity and the foundation of the device. The detailed aptamer functionalization process is described in the methods section. X-ray Photoelectron Spectroscopy (XPS) of the aptamer-modified Au electrodes (Supplementary Fig. 3) shows distinct S $2p$ and N $1s$ peaks originating from the backbone of the aptamer, which indicates the presence of aptamer on the surface of Au electrode. Furthermore, EIS on the Au electrode before and after aptamer modification and mercaptohexanol (MCH) backfill (Supplementary Fig. 4a) show that the

impedance of the Au electrode increased after aptamer modification and MCH backfill over a large frequency range. By fitting the EIS results using a Randles circuit, the double layer capacitance at the Au electrode decreased from ~14.1 μF/cm² to ~8.4 μF/cm² after aptamer modification and further to ~1.7 μF/cm² after MCH backfill (Supplementary Fig. 4b). The EIS results further confirm the successful attachment of aptamer and MCH on the Au electrode. Last, we use EQCM-D to monitor the equivalent functionalization process via mass and electrochemical signal change of the Au-coated QCM sensor (Supplementary Fig. 5). In Supplementary Fig. 5b and c, a large mass increase together with a growing redox peak due to MB (attached on the 5′ end of the aptamer) in stage 1 indicates the bonding and non-covalent absorption process of aptamer on Au surface. The subsequent mass decrease as well as diminishing of the redox peak in the rising step in stage 2 suggest the removal of the non-bonded aptamer. Next, mass increases again in stage 3, owing to the attachment of MCH, and background signal in SWV results decreases, due to MCH's ability to insulate the void areas of the Au electrode. Finally, the last rinsing step partially removes loosely bound MCH. Combining the results from XPS, EIS and EQCM-D, we can confirm the successful aptamer functionalization on the Au electrode, which is critical for our ref-OECT-based E-AB sensor. By using this established aptamer modification protocol[36], the aptamer-modified Au electrode exhibits stable electrochemical behavior indicated by the consistent SWV results over 30 scans (Supplementary Fig. 6).

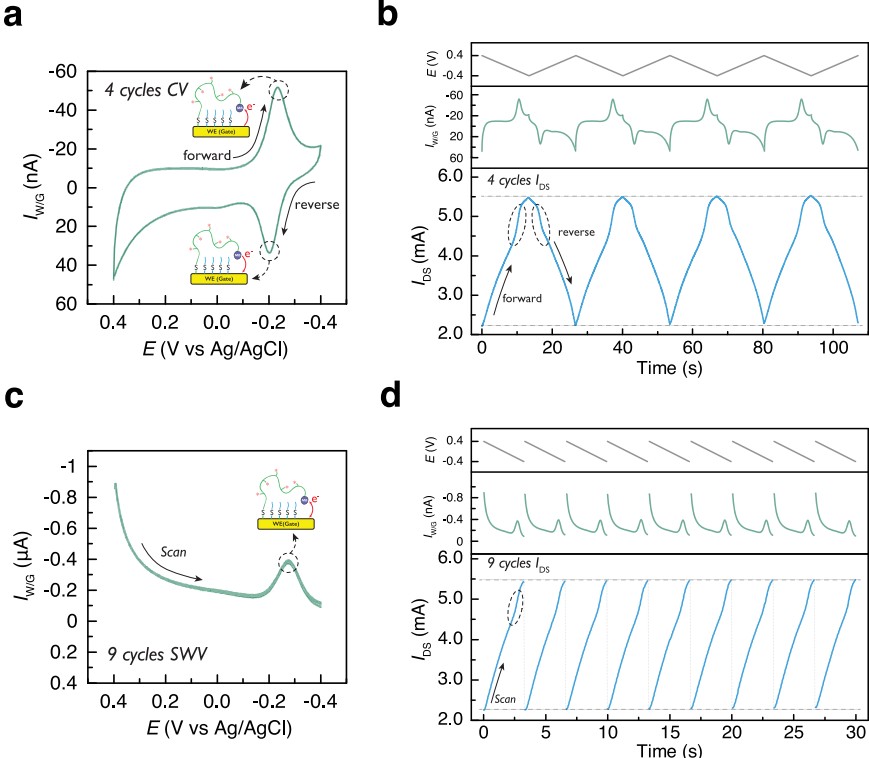

**Fig. 2 | Continuously operation of ref-OECT-based E-AB sensor. a** 4 scans of cyclic voltammetry (CV) of the E-AB sensor with scan rate 60 mV/s and step voltage 0.004 V. **b** Real-time voltage on working electrode, working electrode current/gate current and channel current of the OECT when performing CV measurement in the E-AB sensor. Stable $I_{DS}$ modulation ($V_{DS} = -0.2$ V) is demonstrated during 4 CV scans and the sudden change of $I_{DS}$ indicated by the circles owing to the high reduction/oxidation gate current from the MB redox reporter. **c** 9 scans of square-wave voltammetry (SWV) of the E-AB sensor with scan rate 60 Hz, pulse amplitude 0.04 V and step voltage 0.004 V. **d** Real-time voltage on working electrode (staircase pattern not shown for clarity), working electrode current/gate current and channel current of the OECT when performing SWV measurement in the E-AB sensor. Stable $I_{DS}$ modulation ($V_{DS} = -0.2$ V) is demonstrated during 9 SWV scans and the sudden change of $I_{DS}$ indicated by the circle owing to the high reduction gate current from the MB redox reporter.

One advantage of our ref-OECT-based E-AB sensor is the monolithic integration of various components in a thin film form-factor, which eliminates the use of bulky reference/counter electrodes and enable future miniaturization and system integration. Functionality of the on-chip Ag/AgCl electrode and PEDOT:PSS counter electrode hence needs to be verified before operating the ref-OECT-based E-AB sensor. EIS of an Au electrode was conducted using either bulky Ag/AgCl pellet or on-chip Ag/AgCl as reference electrode (Supplementary Fig. 7a), which shows similar results. Then, by using on-chip Ag/AgCl as a reference electrode, SWV of the aptamer-modified Au was also performed with either a Pt mesh or thin film PEDOT:PSS as the counter electrode, as shown in Supplementary Fig. 7b. Comparable results were also obtained, which confirm the applicability of PEDOT:PSS as the counter electrode.

### Influence of experimental parameters on the electrical behavior of ref-OECT-based E-AB sensor

With the aptamer-modified Au electrode, functional on-chip Ag/AgCl reference electrode and PEDOT:PSS counter electrode, we can demonstrate the operation of ref-OECT-based E-AB sensor. We begin with demonstrating how various parameters influence the behavior of our ref-OECT-based E-AB sensor. The detailed voltage waveform and the sampling strategy of the current is schematically shown in Supplementary Fig. 8. Ref-OECT-based E-AB sensor was first tested by measuring channel current with different $V_{DS}$ while CV/SWV is conducted on the working electrode. As shown in Supplementary Fig. 9, the current in the working electrode remains constant, as expected, in both CV and SWV regardless of the $V_{DS}$ used; while both the channel current and the peak slope of channel current increased with larger $V_{DS}$, indicating that higher signal could be detected at higher $V_{DS}$. We

also evaluate how the scan speed, a crucial parameter for CV experiment, influence the behavior of ref-OECT-based E-AB sensor as shown in Supplementary Fig. 10. As expected, for the CV results in electrode-based sensor, both the amplitude and the separation of oxidation and reduction peaks for MB increase with faster scan rate. For the OECT, the original channel current is recorded as a function of time (only the channel current during reduction scan of MB redox reporter is shown here.) and when the channel current is plotted vs. time, quicker change of channel current as well as higher peak slope can be observed with faster scan speed as expected. When converting the time to voltage by dividing the scan speed and plotting the channel current vs. $V_{g,int}$, scan rate of CV does not influence the channel current or the slope of channel current too much. The peak position of the slope of channel current does show a similar negative shift as that for the reduction peak of electrode. In addition, the stability of ref-OECT-based E-AB sensor was evaluated during continuous operation as shown in Fig. 2. The CV results clearly show 4 overlapped reduction peaks of MB in forward scan and the oxidation peaks of MB in reverse scan in Fig. 2a. As for the OECT measurement in Fig. 2b, the reversibly modulation of the channel current can be observed during the forward and reverse scan of the CV. More importantly, the sudden change of slope in channel current modulation can be observed during both reduction/oxidation process of MB in working electrode in 4 CV cycles, confirm the stably operation of our ref-OECT-based E-AB sensor. In a similar manner to CV, the SWV and OECT results in Fig. 2c and d also show the conversion of redox peak into the slope change of channel current modulation. The similar channel current modulation during 9 SWV scans also indicate the stability of the ref-OECT-based E-AB sensor.

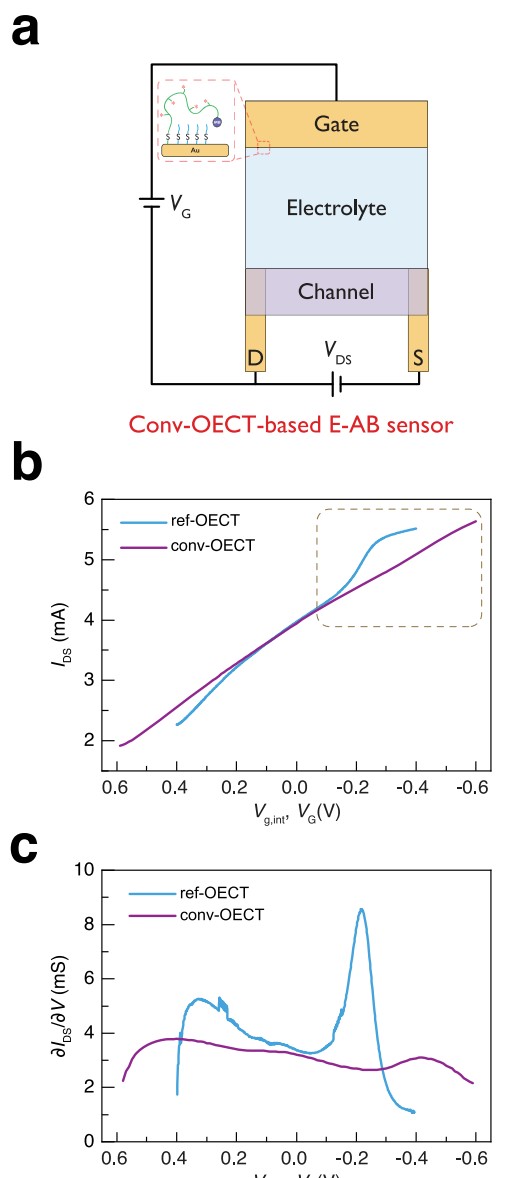

**a**

Conv-OECT-based E-AB sensor

**b**

**c**

**Fig. 3 | Conv-OECT vs. ref-OECT-based E-AB sensor. a** Testing scheme of the conv-OECT-based E-AB sensor. **b, c** Comparison of the channel current and slope of the channel current between ref-OECT and conv-OECT-based E-AB sensor. The scan speed is 60 mV/s for both devices and $V_{DS}$ is −0.2 V. Maximum slope of the channel current is much high in ref-OECT than that in conv-OECT-based E-AB sensor. $V_{g,int}$ describe the voltage drop at gate/electrolyte interface.

## Comparison between ref-OECT and conv-OECT-based E-AB sensor

Before using the ref-OECT-based E-AB sensor for TGF-$\beta_1$ sensing, it is necessary to compare it to conv-OECT-based E-AB sensor (Fig. 3a), which is routinely reported in the literature. Since the most important difference between these two devices (ref-OECT vs. conv-OECT) is the testing method for the working electrode (gate) in the ionic circuit (independent reference electrode has been used in ref-OECT), we first characterize the aptamer-modified Au working electrode in a standard three electrode setup (3E, on-chip Ag/AgCl electrode as reference electrode and PEDOT:PSS channel as counter electrode) and two electrode setup (2E, PEDOT:PSS channel as both reference and counter electrodes). The Au working electrode is exactly the same in these two setups. The results in Supplementary Fig. 11 shows that the redox peak current in the 2E setup is much lower than that of the 3E setup in both

CV and SWV, consistent with previous reports[38,39]. This phenomenon is likely because the evolution of the potential at the working electrode/electrolyte interface is not rigorously controlled in the 2E setup. This allows potential pinning due to the redox reaction which subsequently broadens and flattens the peak (when plotted against simple applied voltage across the entire device), but the total charge associated with the redox process is roughly equivalent in both 3E and 2E setups (Supplementary Fig. 12). Because the modulation of this peak height by aptamer-target binding is the main mechanism for the sensing, it follows that the ref-OECT should provide better sensitivity than conv-OECT-based E-AB sensor. To demonstrate this, OECT channel current in both ref-OECT and conv-OECT-based E-AB sensor with the exact same working electrode (gate of OECT) were measured as shown in Fig. 3b. The conv-OECT-based E-AB sensor operates in a traditional manner where the transfer curve is obtained by scanning the gate voltage ($V_G$) and recording the channel current ($I_{DS}$). While for ref-OECT-based E-AB sensor, CV (only forward scan here) is conducted at the working electrode (gate) in a 3 electrode setup and the channel current is monitored, while simultaneously acting as the counter electrode. Accordingly, the slope of the channel current shows a sudden increase in ref-OECT-based E-AB sensor at the potential associated with the MB charge transfer, while this sudden slope change is significantly diminished in conv-OECT-based E-AB sensor even at a wider scan range. (Fig. 3b). By plotting the derivative of current vs. voltage (slope), it is clear that the peak of the slope is much lower in conv-OECT than ref-OECT-based E-AB sensor (Fig. 3c). This is direct evidence that the redox peak information of MB, which is related to the analyte sensing, can be more effectively converted into the channel current modulation in ref-OECT than that in conv-OECT-based E-AB sensor. Notably, while the configuration used in Fig. 1b is helpful for explaining the mechanism, it should be noted that it is also possible to further simplify the structure of ref-OECT-based E-AB sensor by combining the S and CE contact (Supplementary Fig. 13)".

## TGF-$\beta_1$ sensing with different devices

Next, we show how our ref-OECT-based E-AB sensor improves the sensing ability compared to the electrode-only E-AB sensor. TGF-$\beta_1$ of varying concentration was added into the electrolyte and the CV, SWV of the E-AB sensor, as well as the corresponding OECT channel current modulation (ref-OECT) were recorded simultaneously (Fig. 4). Figure 4a and d clearly show that the increased concentration of TGF-$\beta_1$ results in a decrease of the redox peak in both CV and SWV, which is coherent with the fact that the binding between aptamer and TGF-$\beta_1$ brings the MB away from the electrode surface and decreases the electron transfer rate. As expected, the decreased redox current in CV and SWV due to the TGF-$\beta_1$ also results in a smaller channel current modulation of the ref-OECT-based E-AB sensor as shown in Fig. 4b and e. Calibration curves can be established when TGF-$\beta_1$ concentration is correlated with the redox current in CV (peak-to-peak current)/SWV (peak current) and the channel current modulation in OECT as shown in Fig. 4c and f. When comparing the sensitivity of electrode-based E-AB sensor and the ref-OECT-based E-AB sensor for TGF-$\beta_1$ sensing, ref-OECT-based E-AB sensor shows a ~12000-fold enhancement in sensitivity (290 μA/dec) than CV sensing (24 nA/dec) and ~3500-fold enhancement in sensitivity (292 μA/dec) than SWV sensing (84 nA/dec) in electrode-based E-AB sensor. In order to demonstrate that the decrease of channel current modulation is indeed caused by the binding between TGF-$\beta_1$ and aptamer rather than the degradation of the PEDOT:PSS channel, stability of the PEDOT:PSS channel was also verified during TGF-$\beta_1$ sensing process (Supplementary Fig. 14). Standard OECT measurement was performed every time after introducing the TGF-$\beta_1$ with a new concentration by using on-chip Ag/AgCl electrode as the gate. The transfer curves overlapped well with negligible drift, which confirms the stability of PEDOT:PSS channel and further proves that

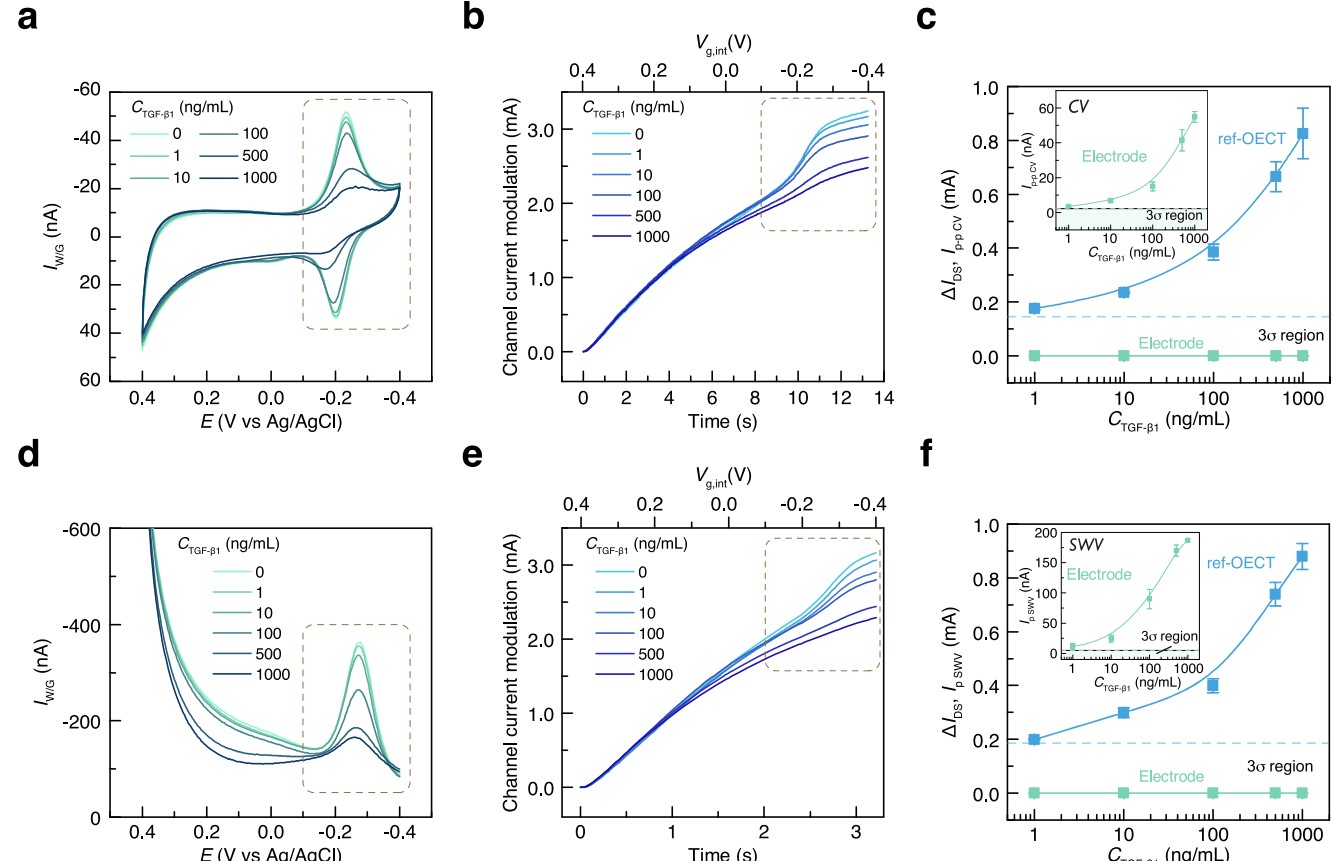

**Fig. 4 | TGF-β₁ sensing. a, d** Series of CV and SWV results of the E-AB sensor with different concentration of TGF-β₁. Decreased redox peak is induced by increasing the concentration of TGF-β₁. **b, e** Series of channel current modulation in ref-OECT with different concentration of TGF-β₁ when performing the CV/SWV measurement in E-AB sensor. Decreased channel current modulation is induced by increasing the concentration of TGF-β₁. **c, f** Calibration curves of TGF-β₁ sensing based on bare E-AB sensors and ref-OECT-based E-AB sensors (error bars represent standard deviation, *N* = 3). Ref-OECT-based E-AB sensor shows ~12000-fold enhancement in

sensitivity (290 µA/dec) than CV sensing (24 nA/dec) and ~3500-fold enhancement in sensitivity (292 µA/dec) than SWV sensing (84 nA/dec) in bare E-AB sensor, with similar detection limit (~1 ng/mL). The $\triangle I_{DS}$ of ref-OECT-based E-AB sensor is extracted at $V_{g,int}$ equal to −0.4 V. The sensitivity is derived from a linear fitting when the concentration of TGF-β₁ is larger than 10 ng/mL. 3σ region is defined by the 3-fold of the standard deviation when performing the CV, SWV and the ref-OECT measurement in bare 1XPBS solution for 10 times.

the decrease of channel current modulation in ref-OECT-based E-AB sensor during TGF-β₁ sensing is indeed caused by the target binding. We also compare the sensing results from conv-OECT and ref-OECT-based E-AB sensor (Supplementary Fig. 15). For conv-OECT-based E-AB sensor, transfer curves were measured as shown in Supplementary Fig. 14a. The calibration curves of the sensors are obtained by plotting the change of the peak height of slope for the channel current as a function of TGF-β₁ concentration. By comparing the calibration curves between conv-OECT and ref-OECT-based E-AB sensor, the ref-OECT-based E-AB sensor has a sensitivity around 2.90 mS/dec, which is about 6 times larger than that in conv-OECT-based E-AB sensor (0.51 mS/dec). The enhanced sensing ability of ref-OECT over conv-OECT-based E-AB sensor indicate that the device (ref-OECT-based E-AB sensor) fully utilizing the property of the redox reporter have better performance. Finally, the selectivity against relevant protein interference of the ref-OECT-based E-AB sensor was characterized (Supplementary Fig. 16) and the aptamer-analyte binding process was also confirmed by EQCM-D (Supplementary Fig. 17).

## Discussion
One advantage of our ref-OECT-based E-AB sensor is the decoupling of sensing and amplification. The working electrode for sensing operates in an ideal 3-electrode setup and obeys the original sensing mechanism of an electrode-based sensor while the mixed-conducting counter

electrode/channel is purely used to amplify the current signal from the working electrode. In this situation, our platform is not only useful for transitioning E-AB sensors to OECT-based sensors, but is also compatible with other electrode-based sensors with various electrochemical interrogation techniques, where the current in the working electrode is used as an indicator for sensing.

Typical OECT-based sensors work in a potential-driven mode where the gate voltage is kept as a constant value or scanned in a range and the channel current is monitored, while gate current is not analyzed as critically. However, in electrochemical sensing, what really matters is the voltage that has been applied at the gate/electrolyte interface, which drives the electrochemical reaction. However, this voltage at the gate/electrolyte interface is usually unknown in conv-OECT-based sensors. While in ref-OECT-based E-AB sensor, the introduction of an additional Ag/AgCl electrode serves as an indicator and helps to control the potential drop at the gate/electrolyte interface where the reaction occurs, while the real voltage applied at the channel from the gate is unknown. However, because of the current continuity from working to counter electrode, the gate current is known during the measurement and can modulate the channel current according to the following equation (see Supplementary note 1)[40]:

$$\triangle I_{DS} = \frac{\mu V_{DS}}{L^2} \int I_G \mathrm{dt} \qquad (1)$$

Where $\triangle I_{DS}$ is the channel current modulation, $L$ is the channel length, $\mu$ is the hole mobility of the mixed conductor and $V_{DS}$ is the drain/source voltage. The integral of $I_G$ stands for the number of injected ions into the mixed conductor that modulates the carrier concentration during operation. In this scenario, our ref-OECT-based E-AB sensor is a current-driven OECT where the detection of the targets will influence $I_G$ and its integral, hence the channel current modulation. Operation of the OECT in galvanostatic mode with constant gate current further support our claim that the modulation of channel current scales linearly with the amount of injected charge ($Q$) in the region of operation of interest (Supplementary Fig. 18). The functionality of Eq. 1 also helps to explain the observed signals in Figs. 3 and 4. Specifically, the electrode E-AB sensor measured by the CV and SWV show the redox peaks associated with the current in working electrode, equivalent to $I_G$ in the transistor. The transduced signal via the OECT then takes on the functionality of the integral of $I_G$ modified by an amplification factor of $\frac{\mu V_{DS}}{L^2}$. Although the integral of $I_G$ is not used to characterize the electrode E-AB sensor, it is positively correlated with the redox current in CV (peak-to-peak current)/ SWV (peak current). As a result, the amplification factor can still be optimized according to Eq. 1. This equation describes the amplification properties of an OECT to the integral of $I_G$, which complements the OECT amplification work of others. For example, the work by Braendlein et al. reports on the use of OECT to amplify input voltage in a voltage amplifier circuit with load resistance[41], and Bonafe et al. shows OECT amplification of AC current signal in the capacitive-dominated regime[42]. The equation also shows that the amplification factor is highly related to the materials figure-of-merit and the geometry, which indicates that we can further enhance the amplification factor by using organic mixed ionic–electronic conductors (OMIECs) active channel with higher mobility and designing the channel with short length. Although volumetric capacitance ($C^*$), another important parameter of OMIECs, is not explicitly shown in Eq. 1, it is still critical for the sensor design. When a channel with low volume is used for miniaturization purpose, a larger $C^*$ is necessary to maintain the charge injection capacity of the channel, and hence its functionality as a counter electrode. In addition, channel material design rules of the ref-OECT-based E-AB sensor are such that the operation region of the OECT must be matched with the redox potential of the redox reporter, among other considerations. For example, accumulation mode OECT with p-type channel can work with a MB redox reporter, while accumulation mode OECTs with n-type channels are compatible with a ferrocene redox reporter.

Finally, one advantage of conventional OECT-based sensors is the removal of the reference electrode that enables simplicity. While in ref-OECT-based E-AB sensor, although we retain the reference electrode, the thin film form-factor and the monolithic integration enable minimal added burden in the device design, which still results in a sensor with minimal size, integration flexibility, and ease of operation. More importantly, multiple sensing gates that target different analytes with shared Ag/AgCl reference electrode and PEDOT:PSS counter electrode/channel can be fabricated, which could lead to the multiplexed sensing and amplification on-site.

In summary, we successfully demonstrated the fabrication and utilization of a ref-OECT-based E-AB sensor that shows 3-4 orders of magnitude enhancement in sensitivity for TGF-$\beta_1$ sensing compared to traditional electrode-based E-AB sensors, as well as improvement in sensitivity compared to conv-OECT-based E-AB sensors. Monolithic integration of aptamer-modified Au working electrode, on-chip Ag/AgCl reference electrode, and PEDOT:PSS counter electrode enables the compact design of the device that has great potential for high density sensing arrays. A new device testing scheme helps to decouple the sensing from E-AB sensor and amplification in OECT, which helps to retain the key features for both OECT and E-AB sensor. This approach guarantees the functionality of both E-AB sensor and OECT and offers an integration strategy between electrode-based sensor with traditional electrochemical characterization methods and OECT. Direct amplification of the current in the working electrode (gate, E-AB sensor) can be achieved on-site in the OECT, reflected by the channel current modulation. This device concept and testing scheme is believed to be universal for E-AB sensors targeting other analytes, as well as other tethered redox-reporter based sensors, and harnesses well known electrochemical interrogation methods while enhancing sensitivity. In addition, it also allows us to integrate previously discussed high surface area sensing electrodes[14–16], and could further enhance both sensitivity and limit of detection (LOD). As such, these separate techniques are not mutually exclusive, and can be integrated synergistically. We believe this approach can enable compact sensing arrays of aptamer-based sensors that are compatible with various targets, which is critical for in vitro and in vivo high throughput diagnostics.

## Methods

### Aptamer preparation

TGF-$\beta_1$ aptamer was purchased from Integrated DNA Technologies (IDT) with amino modification at the 5′ end and thiol modification at the 3′ end. The sequence is shown as:5′/5AmMC6/ CG*CTCGG*CTTC*ACG*AG*ATT*CGTGT*CGTTGTGT*C*CTGT*A*C *C*CG*C*CTTG*A*C*C*AGT*C*ACT*CT*AG*AGC*AT*C*CGG*A*CTG/ iSpC3//3ThioMC3-D/3′[36]. The backbone of the aptamers was partially modified by phosphorothioate bond (represented by "*" in the sequence) on 5′ end of both A and C. This modification is believed to provide enhanced nuclease resistance and higher affinity than the native phosphodiester bond[43]. The aptamer is reconstituted at a concentration of 100 μM in IDTE buffer (pH=8.0) from the supplier. Methylene blue (MB), carboxylic acid, succinimidyl was purchased from Biosearch Technologies. NHS-labeled MB was conjugated to the 5′ end of TGF-$\beta_1$ aptamer via the succinimide ester coupling reported previously[37]. In short, 50 μL of 100 μM aptamer was mixed with 20 μL dimethylformamide (DMF), 10 μL 0.5 M sodium bicarbonate (NaHCO$_3$) and 0.3 mg MB. The mixture was stored at 4 °C for 4 h to modify the aptamer with MB redox reporter. 5 μL MB-modified aptamer was reduced by 10 μL 10 mM tris(2-carboxyethyl)phosphine hydrochloride (TCEP, in IDTE buffer) at room temperature (RT) for 2 h to cleave the disulfer bond in aptamer. This solution was then diluted in 1X Phosphate-buffered saline (PBS) containing 1 mM MgCl$_2$ to 1 μM aptamer concentration and heated at 95 °C for 5 min to re-fold the aptamer. The aptamer solution was ready to be used for modification after cooling down at RT for 15 minutes.

### Fabrication process of ref-OECT-based E-AB sensor

The detailed fabrication process is shown in Supplementary Fig. 1. First, Cr (5 nm)/Au (100 nm) electrodes were patterned on glass substrates by photolithography and the subsequent lift-off process of the negative photoresist (AZ nLOF 2035). After a surface treatment with the adhesion promoter Silane A-174, a 2 μm-thick parylene C was deposited on the substrate serving as an encapsulation layer. Then, a diluted micro-90 (2% v/v in DI water) was spin-coated as an anti-adhesive layer, and subsequently, a sacrificial second parylene C layer of 2 μm was deposited. One of the Au electrodes (2mm × 2mm) was opened through successive photolithography (AZ P4620 photoresist) and reactive ion etching steps (Samco RIE-10NR). 200 nm of Ag was deposited by e-beam evaporator (AJA) and patterned by peeling the sacrificial parylene layer. To form the on-chip Ag/AgCl reference electrode, 0.1 M FeCl$_3$ solution was dropped on the patterned Ag electrode for 1 minute to partially transfer Ag to AgCl. After the fabrication of on-chip Ag/AgCl electrode, PEDOT:PSS counter electrode (channel) was patterned using

similar sacrificial parylene peel-off process. After opening the exposed counter electrode area, PEDOT:PSS blend consisting of 5 vol% ethylene glycol (EG), 1 vol% (3-glycidyloxypropyl) trimethoxysilane (GOPS), and 0.5 vol% dodecylbenzene sulfonic acid (DBSA), filtered through a 0.45 µm polytetrafluoroethylene filter was spin-coated on the device at 2000 rpm for 1 minute. After a gentle thermal annealing at 90 °C for 2 minutes, the sacrificial parylene layer was removed to pattern the PEDOT:PSS counter electrode (channel), followed by thermal crosslinking at 140 °C for 60 min. The last step of the device fabrication was the modification of the 3 sensing electrodes using TGF-$\beta_1$ aptamer. A sacrificial parylene deposition and etching process was performed to open the 3 Au working electrodes, while keeping the other part in the device encapsulated to avoid the influence of aptamer on the other components of the device. The device was then immersed in the prepared aptamer solution for 18 h at 4 °C. After rinsing the unbonded aptamer with DI water, the device was then incubated in 2 mM mercaptohexanol (MCH) solution (in 1XPBS with 1 mM MgCl$_2$) for 3 h at RT. The device was ready to use after rinsing with DI water and peeling off the sacrificial parylene layer.

### X-ray photoelectron spectroscopy (XPS)

The XPS spectrums of aptamer-modified Au electrode were taken using Thermo Scientific ESCALAB 250Xi equipped with a monochromatic KR Al X-ray source (spot size around 500 µm) in Northwestern University's Atomic and Nanoscale Characterization Experimental Center (NUANCE). A flood gun was used for charge compensation. The analysis of the spectrum was performed using the Avantage (Thermo Scientific) software.

### Electrochemical quartz crystal microbalance with dissipation (EQCM-D)

EQCM-D was performed using an Ivium potentiostat connected with a QSense electrochemistry module. Three-electrode setup comprised an Ag/AgCl reference electrode, Pt counter electrode, and the EQCM chip (Quartz PRO, 5.000 MHz, 14 mm Ti/Au) as the working electrode. The mass change was modeled with Sauerbrey equation[44].

### Electrical Characterization

All the electrochemical measurements (EIS, CV, SWV) were conducted using an Ivium potentiostat. OECT channel current was measured by Keithley 2604B source meter with custom-made LabVIEW programs.

## Data availability

The data that support the findings of this study are available from the corresponding author upon request.

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

## Acknowledgements
This research is sponsored by the Defense Advanced Research Projects Agency (DARPA) through Cooperative Agreement D20AC00002 awarded by the U.S. Department of the Interior (DOI), Interior Business Center. The content of the information does not necessarily reflect the position or the policy of the Government, and no official endorsement should be inferred. This work made use of the NUFAB and EPIC facility of Northwestern University's NUANCE Center, which has received support from the ShyNE Resource (NSF ECCS-2025633), the IIN, and Northwestern's MRSEC program (NSF DMR-1720139). We also thank Dr. Bryan Paulsen for the fruitful discussion.

## Author contributions
X.J. and J.R. conceived and oversaw the project. X.J. performed the experiment. X.J. and X.L. analyzed the data. X.J. and J.R. wrote the manuscript. All authors revised the manuscript.

## Competing interests
The authors declare no competing interests.
