## [Peer Review File · Nature Communications]

Organic Electrochemical Transistor as an On-site Signal Amplifier for Electrochemical Aptamer-based SensingREVIEWER COMMENTS

Reviewer #1 (Remarks to the Author):

The manuscript of Xudong Ji et al (Manuscript Nr.: NCOMMS-22-35587) reports on the development of an organic electrochemical transistor system that uses an aptamer receptor to implement a sensor for the detection of the biomarker transforming growth factor beta 1. This sensor contains beside gate, source, and drain electrode also an on-chip reference and counter electrode. The on-chip reference electrode facilitates the exact determination of the electrochemical potential of the gate electrode, however the function of the additional counter electrode remains unclear. The authors mention that they have realized a new "3E-OECT"-based E-AB sensor that retains the features of both OECT and E-AB sensor. However, in my opinion, this would be possible also by conventional "2E-OECT". The main novelty is the systematic investigation of the contribution of the gate redox current on the modulation of the source / drain channel current and the demonstration that it has a considerable impact although the number of redox groups on the gate surface is very small. However, the integration of on-chip reference and counter electrodes is routinely done in the development of electrophysiological and chemical sensor devices.

Overall, the manuscript reports on an experimental work that contains most parts required for a comprehensive study. The characterisation of the devices and the sensor performance were performed technically correct. Nevertheless, the discussion of 2E versus 3E devices comprises major weaknesses and some parts of a complete characterization of the sensor performance are missing. Otherwise, the obtained data mainly support the claimed findings. Given the fact that OECT based aptamer sensors have been reported previously and that E-AB sensors are just one subclass of aptasensors, I believe that the overall significance of the reported work is limited. Consequently, I can not recommend the publication of manuscript under review in this journal.

Please address the following annotations for a publication elsewhere:

Major criticism:

The motivation for the implementation of an E-AB requires a more detailed elaboration. Manifold transducer concepts have been introduced, what makes this type so special? For instance, it was not reasoned why it would be better to measure the change of the charge transfer resistance instead of modulation of the capacitance? Typically, the E-AB signals are small (see nA signals) due to the low number of redox probes (here MB) immobilized on the surface and specific pulsed techniques (SWV) need to be used to access the current signal.

By the way, the claim "large IG" on page 5 line 146 is misleading in this regard.

The function of CE remains unclear. Is the CE and source /drain electrode operated together or separately? Is there a potential difference between CE and source/drain? Why cannot the source or drain electrode take over the function of the CE? In the scheme of Fig.1b it seems as there is a contact

between CE and channel. Is the CE electrode otherwise passivated or is also a part of the free metal exposed to the electrolyte? A schematic would be helpful here.

It remains unclear what the relation between E and V_g is. It makes not much sense to plot these two values above each other since the energy scale is shifted for these two. Both configurations use different reference points, one the Ag/AgCl electrode (defined vs. vacuum level) and the other one the potential of the ground (mainly undefined vs. vacuum level). For the 2E-OECT it remains unclear if the used voltage range covers the redox potential of MB. If this is not the case, then all the observed differences such as the lack of a redox peak in Fig.2b,c could be explained by the fact that the redox potential of MB wasn't reached.

Continuing in this direction, it was actually not explained why there was no redox peak observed for the 2E-system, although the redox process happens on a gold electrode which is (chemically) independent from the channel? The only reasonable explanation is that the redox potential of MB was not reached. However, this could be obtained by shifting the scanned potential range.

The authors claim: "Typical OECT-based sensors work in a potential-driven mode where the gate voltage is kept as a constant value or scanned in a range and the channel current is monitored, while gate current is often ignored." Although the gate current is not always analyzed, usually it is also recorded as "leakage current" and contributes to the variation of the channel characteristics.

"However, this voltage at the gate/electrolyte interface is usually unknown in OECT-based sensors, which also explains the reason why the 2E-OECT-based E-AB sensor cannot capture the redox information from the MB in our measurement." Also here, if the redox potential of MB is passed during sweeping V_g , then a similar influence on the channel characteristics need to be found for the 2E-OECT device since the same redox current flows over the gate electrode.

The increase in sensitivity is not very informative since it does not take the noise level (signal to noise ratio) into account. A better indication of the improved performance is the detection limit since it includes the noise level. Adding an amplifier into the circuit enhances the current signal but also the noise as can be seen from the error bars of the calibration curves.

Typically, not only the sensitivity but also the selectivity and the capability to perform in real samples is tested in the development of sensors.

Annotations of minor importance:

Page 4: "The interdigitated drain and source electrodes are patterned underneath the PEDOT:PSS channel which defines a large W/L to boost the channel current modulation," More information on the channel dimensions are required and how the channel current modulation is boosted.

The authors claimed the transduced signal of the 3E-OECT was related to an amplification factor. According to eq. 1, a higher signal could be detected when higher V_{ds} is applied. Was this experimentally evaluated?

MCH is hardly dissolvable in water. Usually, ethanol or similar solvents are used. How reproducible is this modification step utilizing aqueous solutions?

Please provide real data instead of schematic plots in Fig.S8.

The term "single electrode sensor" is misleading since at least two electrodes (here three electrodes) are required to operate an electrochemical sensor.

Line 350: How do the authors prevent that MB conjugates also with the amino groups of the purin units of the DNA molecules?

The authors should provide the details of buffer solution used.

Figure 4c and f describe the ratio of $\Delta I_{pp}/I_{pp,cv}$ as a function of target concentration with the unit of mA but I assume the unit is not needed. Besides, the insets should be obtained from Figure 4a and 4d and have peak current in the level of μA , which is not consistent with nA shown in a and d.

Reviewer #2 (Remarks to the Author):

This manuscript reported an OECT-based electrochemical aptamer-based (E-AB) sensors made of aptamer modified Au working/gate electrode, on-chip Ag/AgCl reference electrode and a PEDOT:PSS counter electrode/channel. The device architecture couples OECT with standard three-electrode electrochemical system, enabling both OECT-based sensing and conventional electrochemical testing. The design of a PEDOT:PSS counter electrode simultaneously serving as the channel of OECT could amplify the current between the working/ gate electrode and the counter electrode into a significant channel current modulation due to ion doping, endowing the device with high sensitivity. This advantage was demonstrated by using the device to detect transforming growth factor beta 1 (TGF- β 1), which shown higher current response than both conventional E-AB sensors and 2E-OECT-based E-AB sensors and a similar detection limit to that of E-AB sensors. Overall, the device design is novel and interesting, and the features of the device are well explained. The manuscript can be improved if the following questions are discussed and addressed.

1. As discussed in the manuscript (line 275-282), the voltage at the gate/electrolyte interface of the 2E-OECT-based E-AB sensor is different from that applied on the sensing gate electrode of the 3E-OECT-based E-AB sensor, which explains why the reduction peak of methylene blue (MB) was not observed in the 2E-OECT-based E-AB sensor (Figure 2c). The reviewer is very curious about the transfer curves of the 2E-OECT-based E-AB sensors under a wider scan range of gate voltage, which however are not provided. Specifically, for the 2E-OECT-based E-AB sensors, will the reduction peak of MB occur at a different gate voltage?

2. The authors stated that the change of channel current modulation in the 2E-OECT-based E-AB sensor “is likely due to the increased impedance (capacitive) at gate/electrolyte interface upon TGF- β 1 binding, but not the altered charge transfer kinetics from the MB redox reporter” (Line 239-240). This statement should be supported by experimental results (e.g., impedances of the gate electrode before and after TGF- β 1 binding).

3. The selectivity of the 3E-OECT-based E-AB sensor is not well characterized. Can redox active metabolites (e.g., dopamine, uric acid, and ascorbic acid) cause signal response?

4. In Figure 4 c,f, the channel current responses of the sensors were extracted by measuring the transfer curves. Can the sensors be used for real-time monitoring (e.g. IDS-time curve with fixed VDS and $V_{g,int}$ upon adding TGF- β 1 with a series of concentration)? And how about the response speed of the sensor?

5. Will the area of the sensing gate electrode effect the sensitivity of the 3E-OECT-based sensors?

6. It is notably that the absolute current change of the 3E-OECT-based AB sensors is much higher than that of the E-AB sensor, while they show a similar detection limit. Are there any strategies to further improve the detection limit of the 3E-OECT-based E-AB sensors?

7. Comparison of the proposed TGF- β 1 sensor with other techniques used for TGF- β 1 sensing is suggested to be provided to show the advantages of the device.

8. There are many organic mixed ionic–electronic conductors used for OECTs. Why PEDOT:PSS was chosen as the channel materials (counter electrode) in this 3E-OECT-based sensor? Are there any specific considerations in choosing channel materials when design the 3E-OECT-based sensor.

Minor:

9. Please specify the VDS in Figure 2c.

10. The $V_{g,int}$ used for extracting the channel current response shown in Figure 4 c, f should be specified, as the channel current response depends on $V_{g,int}$.

Reviewer #3 (Remarks to the Author):

The manuscript reports on an electrochemical aptamer sensor that exploits OECT based amplification to increase the sensitivity. The device enables superior detection of biomedically relevant marker proteins making the approach very interesting for low-cost point of care sensors. The manuscript gains in novelty as the proposed OECT sensor architecture is reported for the first time and can be generally applied to introduce OECT amplification in amperometric electrochemical sensors. Such a topic is very timely with strong interest from many different research groups. As the contribution is original and well documented I fully support publication of the manuscript in Nat. Commun. However, I have some critical remarks that require a major revision mainly aiming at increasing the clarity of the manuscript and to further support its major claims:

(i) The authors have to address the working mechanism of their OECT enhanced sensor with more detail and clarity. In the current form, the arguments are too simplistic and make it difficult for specialists to understand the work upon first reading. For example in Line 137 the authors argue that “Since the current at the working electrode is equal to the current in counter electrode, which induces the ion injection into the counter electrode that dictates conductivity, we can directly relate the output of the OECT to the current in working electrode in the E-AB sensor.” Although in very general terms the argument is correct, it obscures the main working principle enabling the amplification. In particular, a continuous and constant WE current would cause a continuous change in OECT channel current as explained later. The authors should at least clarify here in the initial part of the manuscript that the OECT counter electrode acts as a capacitively coupled electrode and make reference to the later derivation of the underlying eqn. For readers with a background in electronic engineering it would be helpful to make the reference to a charge sensitive amplifier....

(ii) Considering the proposed operation mechanism, it is also very surprising that the authors distinguish between source and counter electrode. Is there any reason for this? I expect that the lithographic pattern separating source and CE is not necessary. Such a simplification would enormously enhance understanding of the device.

(iii) The OECT-counter electrode approach should be demonstrated with a model experiment in which the potentiostat is operated in galvanostatic mode. Controlled current pulses subjected into the working electrode should then be related to IDS current increase. Overall, the predicted linear relation between injected Q and measured Δ_{IDS} (eqn.1) should be experimentally demonstrated in this way.

(iv) A crucial parameter for CV experiments is the scan rate. Scan rates used in the described experiments are not reported. How does the scan rate impact on the observed amplification provided by the 3E-OECT device? This point should be discussed and related experimental data should be shown (at least in Suppl. Mat.).

(v) Overall, the comparison with the 2E-OECT is interesting and relevant, given that this configuration is often found in literature. However, I would give priority to a thorough introduction of the new device concept and only later compare 3E-OECT with 2E-OECT as done in figure 3.

(vi) In the discussion section, a quantitative treatment of the amplification is derived. This should be compared with other articles that quantify amplification of OECT based electrochemical sensors, such as Braendlein, M. et al. Adv. Sci. 2017 or Bonafe et al. Nat. Commun. 2022 et al.

Minor points:

Figure 4 c: The y-axis label should be without units.

The variation in OECT drain current is denoted by I_{DS_m} . This is unusual. The author's should use the delta symbol Δ throughout the manuscript to make clear that here a change in current is described.

The introduction makes reference to "the typical sensing mechanism of E-AB sensors". But such a typical mechanism is not described. Is it the variation of an electron transfer rate to a redox moiety in the Aptamer? A better definition should be added to the introduction.

Response to reviewers:

We thank the reviewers for their careful consideration and helpful comments. Below we address their questions and comments, introducing new experiments and describing additions or changes in the text.

Reviewer comments are in black.

Our responses in red.

Additions made to the text are in highlighted red.

Reviewer #1:

The manuscript of Xudong Ji et al (Manuscript Nr.: NCOMMS-22-35587) reports on the development of an organic electrochemical transistor system that uses an aptamer receptor to implement a sensor for the detection of the biomarker transforming growth factor beta 1. This sensor contains beside gate, source, and drain electrode also an on-chip reference and counter electrode. The on-chip reference electrode facilitates the exact determination of the electrochemical potential of the gate electrode; however, the function of the additional counter electrode remains unclear. The authors mention that they have realized a new “3E-OECT”-based E-AB sensor that retains the features of both OECT and E-AB sensor. However, in my opinion, this would be possible also by conventional “2E-OECT”. The main novelty is the systematic investigation of the contribution of the gate redox current on the modulation of the source/drain channel current and the demonstration that it has a considerable impact although the number of redox groups on the gate surface is very small. However, the integration of on-chip reference and counter electrodes is routinely done in the development of electrophysiological and chemical sensor devices. Overall, the manuscript reports on an experimental work that contains most parts required for a comprehensive study. The characterization of the devices and the sensor performance were performed technically correct. Nevertheless, the discussion of 2E versus 3E devices comprises major weaknesses and some parts of a complete characterization of the sensor performance are missing. Otherwise, the obtained data mainly support the claimed findings. Given the fact that OECT based aptamer sensors have been reported previously and that E-AB sensors are just one subclass of aptasensors, I believe that the overall significance of the reported work is limited. Consequently, I cannot recommend the publication of manuscript under review in this journal. Please address the following annotations for a publication elsewhere:

We thank the reviewer’s assessment about “the systematic investigation of the contribution of the gate redox current on the modulation of the source/drain channel current and the demonstration that it has a considerable impact.” However, we do not agree the reviewer’s statement that “the overall significance of the reported work is limited”. While E-AB sensors is but one class of relevance, they are commonly used in the field, with important implementation in vivo. Furthermore, the approach is also relevant for other electrochemical based bioanalyte moieties other than aptamers. In addition, we have conducted additional experiments with more characterization of the 3E-OECT-based E-AB sensor and comprehensively compared the performance of it to the 2E-OECT-based E-AB sensor (commonly implemented). We also characterized the selectivity of the 3E-OECT-based E-AB sensor, and added additional explanations about the function of CE together with other experimental details that the reviewer requested.

Major criticism:

1. The motivation for the implementation of an E-AB requires a more detailed elaboration. Manifold transducer concepts have been introduced, what makes this type so special? For instance, it was not reasoned why it would be better to measure the change of the charge transfer resistance instead of modulation of the capacitance? Typically, the E-AB signals are small (see nA signals) due to the low number of redox probes (here MB) immobilized on the surface and specific pulsed techniques (SWV) need to be used to access the current signal. By the way, the claim “large I_G ” on page 5 line 146 is misleading in this regard.

Response: Thanks for your comment. First, we focus on the electrochemical aptamer-based sensor rather than optical aptamer-based sensor due to the opportunity of E-AB sensor for point-of-care usage, miniaturization sensing and potential capability of in-vivo implementation. In terms of the electrochemical transducing approach, we do agree that both change of charge transfer resistance and modulation of capacitance can be used as an indicator for target binding. However, we believe that transducing the charge transfer resistance between a tethered redox reporter and electrode can provide improved sensitivity than the measurement of bare capacitance change, at least in our measurement. We performed the EIS measurement of the Au electrode modified by aptamer with MB redox reporter, and compare the impedance change of the electrode with different concentrations of TGF- β_1 as shown in **Figure R1**. Here, we compare the Impedance change at frequency equal to 1Hz and find that the modulation of impedance is larger when DC offset in the modified Au working electrode is -0.3V instead of 0V. Since the redox potential of MB is around -0.2V~-0.3V vs Ag/AgCl, the change of impedance when the DC offset is -0.3V is mainly attribute to the altered charge transfer resistance between MB and electrode surface (caused by the binding induced conformational change). While the impedance changes when the DC offset is 0V is mostly contributed by the modulation of interfacial capacitance change after target binding. Because of this, we believe it is better to measure the change of charge transfer resistance instead of modulation of the capacitance, at least in our particular case. In addition, E-AB sensors (specifically ones where the electrode is modified with a redox reporter modified aptamer) have gained significant attention, including demonstrations of real-time, in-vivo sensing in the past two decades. As such, we believe the E-AB sensor has promising potential in medical applications for precision diagnostics. As a result, we would like to contribute to the development of this field by introducing a new OECT signal amplification method described in this work.

We agree that SWV is usually used to access the small current signal, and we also show in the main text that the SWV sensing method yields higher sensitivity than CV. However, even compared with SWV technique, our 3E-OECT approach still shows improved sensitivity, with around 3500-fold enhancement.

In terms of I_G , we aim to compare the one before and after target binding, so we use “large” and “small”. To avoid the possibly misleading to the reader, we have changed “large” and “small” to “high” and “low”.

Revision made: Based on the comment from the reviewer, we have added this sentence in page 2 of the revised manuscript as “In addition to the applicability towards various sensing targets, the potential for real-time sensing and in-vivo implementation have motivated E-AB sensor development”. We have also changed the “large” and “small” to “high” and “low” when describing the I_G in working electrode.

Figure R1. EIS results of the MB-tethered aptamer modified Au electrode after challenging by TGF- β_1 with different concentration when (a) $V_{DC}=0V$ and (b) $V_{DC}=-0.3V$. (c) Impedance change at 1Hz for TGF- β_1 with different concentration at $V_{DC}=0V$ or $V_{DC}=-0.3V$.

2. The function of CE remains unclear. Is the CE and source /drain electrode operated together or separately? Is there a potential difference between CE and source/drain? Why cannot the source or drain electrode take over the function of the CE? In the scheme of Fig.1b it seems as there is a contact between CE and channel. Is the CE electrode otherwise passivated or is also a part of the free metal exposed to the electrolyte? A schematic would be helpful here.

Response: We apologize about the confusion here for the function of CE. Actually, patterned PEDOT:PSS layer is used as both OECT channel and counter electrode. S, D and CE labelled in Figure 1a and b are encapsulated by parylene layer except for the area that is directly contact with PEDOT:PSS. The CE annotated in the figure is just the metal line that is used to connect the PEDOT:PSS counter electrode. In response to this comment from the reviewer, we performed an additional experiment to compare the results by either merging the S and CE as one electrode or separate them as two individual contacts. As shown in **Figure R2**, these two testing methods give very similar results in terms of both CV in working electrode and the current in OECT channel (counter electrode). Thus, we believe the source electrode can take over the function of the CE as suggested by the reviewer. Since all other experiments are performed with CE/S separated, and since this wiring can serve to describe the operation of this arrangement, we elect to keep the original configuration in the main text. However, we note that this simplification can be readily made and include the data in the SI.

Revision made: Based on the comment from the reviewer, we have added this sentence in page 8 of the revised manuscript as “Notably, while the configuration used in Fig. 1b is helpful for explaining the mechanism, it should be noted that it is also possible to further simplify the structure of 3E-OECT-based E-AB sensor by combining the S and CE contact (Supplementary Fig. 12).” We also added Supplementary Fig. 12 in Supplementary Information.

Figure R2. Schematic image of the device testing scheme by either (a) separating the S and CE contact or (b) combining S and CE contact. (c) CV results in working electrode with two different testing schemes. (d) Channel current of OECT with two different testing schemes.

3. It remains unclear what the relation between E and V_g is. It makes not much sense to plot these two values above each other since the energy scale is shifted for these two. Both configurations use different reference points, one the Ag/AgCl electrode (defined vs. vacuum level) and the other one the potential of the ground (mainly undefined vs. vacuum level). For the 2E-OECT it remains unclear if the used voltage range covers the redox potential of MB. If this is not the case, then all the observed differences such as the lack of a redox peak in Fig.2b,c could be explained by the fact that the redox potential of MB wasn't reached. Continuing in this direction, it was actually not explained why there was no redox peak observed for the 2E-system, although the redox process happens on a gold electrode which is (chemically) independent from the channel? The only reasonable explanation is that the redox potential of MB was not reached. However, this could be obtained by shifting the scanned potential range.

Response: We thank the reviewer for the insightful comment, which is very helpful for us to better understand the device physics and enable us to make a fairer comparison between our 3E-OECT-based E-AB sensor with the traditional 2E-OECT approach. To address this comment, we performed additional experiments and also added more analysis of our old data to compare the performance of 3E and 2E OECT approach and demonstrated the superiority of the 3E-OECT over 2E-OECT. First of all, we focused on characterizing the aptamer modified

Au working electrode in both standard three electrodes setup (3E, on-chip Ag/AgCl electrode as reference electrode and PEDOT:PSS channel as counter electrode) and two electrodes setup (2E, PEDOT:PSS channel as both reference and counter electrodes). The Au working electrode is exactly the same in these two setups and the device geometry is the same as what we used in the main text. We compare the results of both CV and SWV in working electrode with 3E and 2E setup as shown in **Figure R3**. Indeed, as the reviewer suggests, the MB redox peak is visible in both CV and SWV by using an extended scan range with a peak position slightly over -0.4V. However, compared to the results in 3E setup, the redox peak in 2E setup shows a much lower current in both CV and SWV. Because the modulation of this peak height by aptamer-target binding is the main mechanism for the sensing, it follows that the 3E-OECT should provide better sensitivity compared to 2E-OECT. To validate this, we then compared the OECT channel current in both 3E and 2E-OECT with exactly the same working electrode (now as the gate of the OECT). Because the scan range of V_g is wider in 2E-OECT compared to that in 3E-OECT, comparing the channel current modulation in 2E and 3E OECT (as was originally done in the main manuscript) is not a fair comparison in our point of view. Instead, we compare the difference in the slope (transconductance) in 2E and 3E-OECT. We show the results from two batch of devices, one from early 2021 (**Figure R4a and b**), the other from late 2022 (**Figure R4c and d**). As we can see that for 2E-OECT, with enlarged scan range, the slope of channel current indeed shows an increase at the position of MB redox peak, and the transconductance also show a peak there. While when comparing the channel current in 2E-OECT with 3E-OECT, the peak of the slope (transconductance) is much lower in 2E-OECT than 3E-OECT (**Figure R4b and d**). The results from these two batches of devices agree with the finding that the influence of the MB redox reaction is more prominent in 3E-OECT compared to 2E-OECT. Lastly, we also compared the sensing results from 2E and 3E-OECT as shown in **Figure R5**. The calibration curve of the sensor is obtained by plotting the change of the peak height of slope for the channel current as a function of TGF- β_1 concentration. By comparing the calibration curve between 2E and 3E OECT, the 3E-OECT has a sensitivity around 2.895 mS/dec, which is around 6 times larger than that in 2E-OECT (0.508 mS/dec). With those additional experiments and analysis, we hope we can convince the reviewer that our 3E-OECT approach has better sensing ability than the 2E approach.

Revision made: Based on the comment from the reviewer, we have intensively modified the paragraph from page 7 to page 8 about the comparison between 3E and 2E-OECT-based E-AB sensor. The whole paragraph is revised as: “Before using the 3E-OECT-based E-AB for TGF- β_1 sensing, it’s necessary to compare it to 2E-OECT-based E-AB sensor (**Fig. 3a**), which is routinely reported in the literature. Since the most important difference between those two devices (3E vs. 2E-OECT) is the testing method for the working electrode (gate) in the ionic circuit, we first characterize the aptamer-modified Au working electrode in a standard three electrode setup (3E, on-chip Ag/AgCl electrode as reference electrode and PEDOT:PSS channel as counter electrode) and two electrode setup (2E, PEDOT:PSS channel as both reference and counter electrodes). The Au working electrode is exactly same in these two setups. The results in Supplementary Fig. 11 shows that the redox peak current in 2E setup is much lower than that of the 3E setup in both CV and SWV. Because the modulation of this peak height by aptamer-target binding is the main mechanism for the sensing, it follows that the 3E arrangement should provide better sensitivity than 2E-OECT-based E-AB sensor. To demonstrate this, OECT channel current in both 3E and 2E-OECT-based E-AB sensor with the exact same working electrode (gate of OECT) were measured as shown in Fig. 2b. The 2E-OECT-based E-AB sensor operates in a traditional manner where the transfer curve is obtained by scanning the gate voltage (V_G) and recording the channel current (I_{DS}). While for 3E-OECT-based E-AB sensor, CV (only forward scan here) is conducted in the working electrode (gate)

in 3 electrodes setup and the channel current is monitored, while simultaneously acting as the counter electrode. Accordingly, the slope of the channel current shows a sudden increase in 3E-OECT-based E-AB sensor at the potential associated with the MB charge transfer, while this sudden slope change is significantly diminished in 2E-OECT-based E-AB sensor even at a wider scan range. (Fig. 2b). By plotting the derivative of current vs. voltage (slope), it is clear that the peak of the slope is much lower in 2E than 3E-OECT-based E-AB sensor (Fig. 2c). This is direct evidence that the redox peak information of MB, which is related to the analyte sensing, can be more effectively converted into the channel current modulation in 3E- than that in 2E-OECT-based E-AB sensor. Notably, while the configuration used in Fig. 1b is helpful for explaining the mechanism, it should be noted that it is also possible to further simplify the structure of 3E-OECT-based E-AB sensor by combining the S and CE contact (Supplementary Fig. 12).”

We also revised the comparison of the sensing results between 3E and 2E-OECT-based E-AB sensor in page 9 of the manuscript as: “We also compare the sensing results from 2E and 3E-OECT-based E-AB sensor (Supplementary Fig. 14). For 2E-OECT-based E-AB sensor, transfer curves were measured as shown in Supplementary Fig. 14a. The calibration curves of the sensors are obtained by plotting the change of the peak height of slope for the channel current as a function of TGF- β_1 concentration. By comparing the calibration curves between 2E and 3E-OECT-based E-AB sensor, the 3E-OECT-based E-AB sensor has a sensitivity around 2.90 mS/dec, which is about 6 times larger than that in 2E-OECT-based E-AB sensor (0.51 mS/dec).” We also added Supplementary Fig. 11, 12 and 14 in Supplementary Information, together with the revised Fig. 3 in main text.

Figure R3. Comparison of the MB redox peak in 3E and 2E setup during (a) CV or (b) SWV measurement. For CV, the scan speed is 100 mV/s and the step size is 5 mV for both 2E and 3E setup. For SWV, the scan frequency is 60 Hz, pulse amplitude is 40 mV and step potential is 4 mV for both 2E and 3E setup.

Figure R4. Comparison of the channel current and slope of the channel current between 3E-OECT and 2E-OECT in the device from (a, b) batch-2021 and (c, d) batch-2022. The scan speed is 60 mV/s for the devices (both 2E and 3E) in batch-2021 and 100 mV/s for the devices (both 2E and 3E) in batch-2022. For 2E-OECT, the x-axis is V_G ; for 3E-OECT, the x-axis is $V_{g,int}$.

Figure R5. (a) Transfer curves of 2E-OECT-based E-AB sensor in electrolyte with different concentration of TGF-β₁. (b) Extracted transconductance of 2E-OECT-based E-AB sensor with different TGF-β₁ concentration. (c) Extracted slope of channel current of 3E-OECT-based E-AB sensor with different TGF-β₁ concentration. Original sensing results is from **Figure 4b** in main text. (d) Comparison of the transconductance (2E)/slope of channel current (3E) with respect to the concentration of TGF-β₁. 3E-OECT based E-AB sensor shows a sensitivity of 2.90 mS/dec, which is around 6 times larger than that in 2E-OECT based E-AB sensor (0.51 mS/dec).

4. The authors claim: “Typical OECT-based sensors work in a potential-driven mode where the gate voltage is kept as a constant value or scanned in a range and the channel current is monitored, while gate current is often ignored.” Although the gate current is not always analyzed, usually it is also recorded as “leakage current” and contributes to the variation of the channel characteristics.

Response We agree with the reviewer’s point that the gate current will be recorded as leakage current and contributes to the variation of the channel characteristics. Often in the OECT literature, however, it is not intensively analyzed, here, however it is critical to understanding the signal transduction. We change our statement with a softer tone and replace “...while gate current is often ignored.” to “...while gate current is not analyzed as critically.”

Revision made: Based on the comment from the reviewer, we change the sentence in page 10 from “...while gate current is often ignored.” to “...while gate current is not analyzed as critically.”

5. “However, this voltage at the gate/electrolyte interface is usually unknown in OECT-based sensors, which also explains the reason why the 2E-OECT-based E-AB sensor cannot capture the redox information from the MB in our measurement.” Also here, if the redox potential of MB is passed during sweeping V_g , then a similar influence on the channel characteristics need to be found for the 2E-OECT device since the same redox current flows over the gate electrode.

Response: We thank the reviewer for pointing this out. As we described in the response to the question 3 from the reviewer, the 2E-OECT indeed shows a slope change in transfer curve when the redox potential of MB is passed with wider scan range of V_G . While the peak of this slope (transconductance) is determined by the height of the redox peak of MB but not the total redox current flow (time integral of all the redox current) through the working electrode during scan. As a result, because MB shows higher redox peak in 3E-OECT than 2E-OECT (**Figure R3**), we still see the 3E-OECT has higher sensitivity than 2E-OECT when calibrated by plotting the transconductance (2E), or slope of channel current (3E) with respect to the concentration of TGF- β_1 .

Revision made: Based on the comment from the reviewer and also the fact that the MB peak can be observed in 2E-OECT with extended scan range, we revised our statement as: “However, this voltage at the gate/electrolyte interface is usually unknown in OECT-based sensors with 2E setup.”

6. The increase in sensitivity is not very informative since it does not take the noise level (signal to noise ratio) into account. A better indication of the improved performance is the detection limit since it includes the noise level. Adding an amplifier into the circuit enhances the current signal but also the noise as can be seen from the error bars of the calibration curves.

Response: We agree that adding an amplifier will enhance both noise and signal, so we didn't claim our 3E-OECT-based sensor has better detection limit than electrode-based sensor. For a biosensor, both sensitivity and detection limit are important indicators for the sensor performance and considering our 3E-OECT-based sensor has similar detection limit but much higher sensitivity than electrode-based sensor, it's reasonable to claim that the 3E-OECT-based sensor has better performance than electrode-based sensor. In this sense, it can be considered that the OECT acts as a co-localized amplifier to the gate/working electrode sensor; it enables tight integration of sensor and amplifier in a compact footprint which will be critical for future in vivo implementation. At the same time, since the sensing is happening at the working electrode that is decoupled from the amplification in the OECT channel, it is straightforward to improve the detection limit of 3E-OECT-based sensor by modifying the working electrode, for example, changing the planar Au to nanostructured Au. Because increasing the surface area of working electrode is generally accepted as a method to improve the detection limit of an electrochemical biosensor, we believe it is also adaptable to our 3E-OECT-based sensor to improve detection limit.

7. Typically, not only the sensitivity but also the selectivity and the capability to perform in real samples is tested in the development of sensors.

Response: We agree with the reviewer that the selectivity and the capability for testing in real sample is important for a sensor development. In terms of selectivity, we performed additional experiments by testing our sensor with two different targets, one is TGF- β_3 and the other is IFN- γ . As shown in Figure R6, introducing 1 μ g/mL TGF- β_3 and 2 μ g/mL IFN- γ didn't cause considerable change of both the working electrode current (**Figure R6a**) and the channel

current of 3E-OECT-based E-AB sensor (**Figure R6b**) because of the specificity of the TGF- β_1 aptamer against TGF- β_1 . As for the slope of the channel current in 3E-OECT, both peak position and height remained similar after the introduction of either TGF- β_3 or IFN- γ . These results show good selectivity of our sensor. In terms of the capability for testing in real sample, we believe this work moves us in the right direction for future development of our 3E-OECT based E-AB sensor together with the *in-vivo* implementation. However, this manuscript is mainly focused on the demonstration of this new sensing concept and how it outperforms the traditional 2E-OECT based E-AB sensor and electrode-based E-AB sensor, as such, we believe that testing real samples is beyond the scope of the current work.

Revision made: Based on the comment from the reviewer, we have added the sentence in page 9 as: "...the selectivity against relevant protein interference of the 3E-OECT-based E-AB sensor was characterized (Supplementary Fig. 15)..." We also added Supplementary Fig. 15 in Supplementary Information.

Figure R6. (a) CV results of the aptamer modified Au working electrode in bare 1xPBS buffer solution and the PBS buffer solution with either 2 $\mu\text{g/mL}$ IFN- γ or 2 $\mu\text{g/mL}$ IFN- γ together with 1 $\mu\text{g/mL}$ TGF- β_3 . (b, c) Channel current and slope of channel current of 3E-OECT based E-AB sensor in bare 1xPBS buffer solution and the PBS buffer solution with either 2 $\mu\text{g/mL}$ IFN- γ or 2 $\mu\text{g/mL}$ IFN- γ together with 1 $\mu\text{g/mL}$ TGF- β_3 .

Annotations of minor importance:

8. Page 4: "The interdigitated drain and source electrodes are patterned underneath the PEDOT:PSS channel which defines a large W/L to boost the channel current modulation," More information on the channel dimensions are required and how the channel current modulation is boosted.

Response: We mentioned the W and L in page 4 line 129~130 of the original manuscript, and the W is 1080 μm , L is 20 μm . Cause the modulation of channel current scales with the W/L ratio according to the equation that describe the operation of OECT, and the interdigitated design of source and drain electrode gives us a large W/L as 54, so we claim that this design will boost the channel current modulation.

Revision made: Based on the comment from the reviewer, we have added the W/L value.

9. The authors claimed the transduced signal of the 3E-OECT was related to an amplification factor. According to eq. 1, a higher signal could be detected when higher V_{ds} is applied. Was this experimentally evaluated?

Response: We thank the reviewer for the comments. We have conducted additional experiments and tested the channel current of 3E-OECT with different V_{DS} in both CV-OECT and SWV-OECT operation. As shown in **Figure R7**, the current in the working electrode remained the same in both CV and SWV regardless of the V_{DS} used, as expected, while for the channel current modulation and the peak slope of channel current, both of them increased with larger V_{DS} . Since both these two values can be used in the characterization of sensing results, it indicates that a higher V_{DS} results in larger detected signals.

Revision made: Based on the comment from the reviewer, we have added the sentence in page 7 as: “3E-OECT-based E-AB sensor was first tested by measuring channel current with different V_{DS} while CV/SWV is conducted on the working electrode. As shown in Supplementary Fig. 9, the current in the working electrode remains constant, as expected, in both CV and SWV regardless of the V_{DS} used; while both the channel current and the peak slope of channel current increased with larger V_{DS} , indicating that higher signal could be detected at higher V_{DS} .” We have also added Supplementary Fig. 9 in Supplementary Information.

Figure R7. (a, d) CV and SWV results of the aptamer modified Au working electrode in bare 1xPBS buffer solution with different V_{DS} . (b, e) Channel current of 3E-OECT based E-AB sensor in CV-OECT or SWV-OECT operation with different V_{DS} . (c, f) Slope of channel current of 3E-OECT based E-AB sensor in CV-OECT or SWV-OECT operation with different V_{DS} .

10. MCH is hardly dissolvable in water. Usually, ethanol or similar solvents are used. How reproducible is this modification step utilizing aqueous solutions?

Response: We thank the reviewer for the comments. Indeed, the solubility of MCH in aqueous solution is worse than that in ethanol or other non-aqueous solvent, however, it is still soluble in aqueous solution to some extent. As this manuscript does not focus on new chemistries of the working electrode, we based the process on prior literature: the backfill of the electrode using MCH in aqueous solution is often reported in the literature for example.^{1,2} In addition, in our manuscript, we show the successfully deposition of MCH on Au surface by EIS and

EQCM-D measurement (reported in original SI). As such, we believe it is reasonable to use MCH in aqueous solution for backfilling aptamer-modified electrode, but note that any assembly should be applicable to our OECT amplification strategy.

11. Please provide real data instead of schematic plots in Fig.S8.

Response: We thank the reviewer for the comments. For Fig. S8, we want to show how we sample the current during CV-OECT and SWV-OECT measurement. Because the CV and SWV results in working electrode is already processed and displayed in the software of the potentiostat, we just show how we sample the channel current of OECT in the revised Fig. S8 for simplicity. The real data is provided as shown in **Figure R8**.

Revision made: : Based on the comment from the reviewer, we have added Supplementary Fig. 8 in Supplementary Information.

Figure R8. (a) Voltage waveform of CV measurement (voltage is ramped up stepwise instead of true linear). (b, c) Raw data and smoothed data of the channel current in 3E-OECT-based E-AB sensor during CV-OECT operation. (d) Voltage waveform of SWV measurement. (e, f) Raw data and sampled data of the channel current in 3E-OECT based E-AB sensor during SWV-OECT operation.

12. The term "single electrode sensor" is misleading since at least two electrodes (here three electrodes) are required to operate an electrochemical sensor.

Response: We thank the reviewer for the comments. We agree that this nomenclature could be misleading, so we change the "single electrode sensor" to "electrode-based sensor".

Revision made: Based on the comment from the reviewer, we have changed all the "single electrode-based sensor" to "electrode-based sensor"

13. Line 350: How do the authors prevent that MB conjugates also with the amino groups of

the purin units of the DNA molecules? The authors should provide the details of buffer solution used.

Response: We thank the reviewer for the comments. We do agree that both DNA backbone and terminal has amine that can be conjugated with NHS-labelled MB through succinimide ester coupling reaction; while this reaction occurs via the free amines, which means that the conjugation efficiency is pH dependent considering the protonation status of the amine in buffer with different pH. The basicity of the DNA base amine ($pK_b \sim 9.5$) is significantly different than the terminal amine ($pK_b \sim 4$).³ As a result, since the buffer solution we use during the MB labelling process is 0.5 M NaHCO_3 with pH 8, it is reasonable to predict that the coupling efficiency for the terminal amine is better than DNA base amine. In addition, as mentioned above, we based these protocols on previously reported works: there are a few papers using the same protocol to label the MB on the TGF- β_1 aptamer for sensing.⁴ Again, we do not intend to report new methodology for the functionalization of the working/sensing electrode in this work, however, these are important questions.

14. Figure 4c and f describe the ratio of $\Delta I_{DS}/I_{pp,CV}$ as a function of target concentration with the unit of mA but I assume the unit is not needed. Besides, the insets should be obtained from Figure 4a and 4d and have peak current in the level of μA , which is not consistent with nA shown in a and d.

Response: We apologize for the misleading figure. The y axis in Figure 4c does not describe the ratio between ΔI_{DS} and $I_{pp,CV}$, but the real value of either ΔI_{DS} for 3E-OECT OR $I_{pp,CV}$ for working electrode. For the inset of Figure 4C, because it is enlarged from Figure 4C, the unit is actually $\mu \times \text{mA}$, which is nA. To make it clear, we change the unit in the inset from μ to nA.

Revision made: Based on the comment from the reviewer, we have changed the unit in the inset of both Fig. 4c and f from μ to nA.

Reviewer #2:

This manuscript reported an OECT-based electrochemical aptamer-based (E-AB) sensors made of aptamer modified Au working/gate electrode, on-chip Ag/AgCl reference electrode and a PEDOT:PSS counter electrode/channel. The device architecture couples OECT with standard three-electrode electrochemical system, enabling both OECT-based sensing and conventional electrochemical testing. The design of a PEDOT:PSS counter electrode simultaneously serving as the channel of OECT could amplify the current between the working/gate electrode and the counter electrode into a significant channel current modulation due to ion doping, endowing the device with high sensitivity. This advantage was demonstrated by using the device to detect transforming growth factor beta 1 (TGF- β_1), which shown higher current response than both conventional E-AB sensors and 2E-OECT-based E-AB sensors and a similar detection limit to that of E-AB sensors. Overall, the device design is novel and interesting, and the features of the device are well explained. The manuscript can be improved if the following questions are discussed and addressed.

We thank the reviewer to find "...the device design is novel and interesting, and the features of the device are well explained." We performed additional experiments and hope the reviewer's concern can be addressed by the supplementary information provided.

1. As discussed in the manuscript (line 275-282), the voltage at the gate/electrolyte interface of the 2E-OECT-based E-AB sensor is different from that applied on the sensing gate electrode of the 3E-OECT-based E-AB sensor, which explains why the reduction peak of methylene blue (MB) was not observed in the 2E-OECT-based E-AB sensor (Figure 2c). The reviewer is very curious about the transfer curves of the 2E-OECT-based E-AB sensors under a wider scan range of gate voltage, which however are not provided. Specifically, for the 2E-OECT-based E-AB sensors, will the reduction peak of MB occur at a different gate voltage?

Response We thank the reviewer for the comment. This comment is similar to comment 3 of reviewer 1. Please also refer the answer. In short, we do see the effect of MB redox peak on the transfer curve of 2E-OECT based E-AB sensor with wider scan range of V_G , while the slope change in transfer curve of 2E-OECT is much smaller than that in 3E-OECT as shown in **Figure R4**. In addition, we also measured the transfer curve of 2E-OECT in a wider V_G scan range with different V_{DS} as shown in **Figure R9**. The effect of MB redox peak on the transfer curve can be observed in all V_{DS} used, as well as a negative shift of peak position of the transconductance ($\partial I_{DS}/\partial V_G$) with increased V_{DS} . For comparison, the peak slope of channel current in 3E-OECT is much higher than that in 2E-OECT regardless of the V_{DS} used (**Figure R7c**). At the same time, the peak position of the slope of channel current in 3E-OECT stays approximately the same with different V_{DS} .

Revision made: Please refer to the revision made as requested by reviewer 1 in comment 3.

Figure R9. (a, b) Transfer curves and transconductance of 2E-OECT based E-AB sensor with wider scan range of V_G and different V_{DS} .

2. The authors stated that the change of channel current modulation in the 2E-OECT-based E-AB sensor "is likely due to the increased impedance (capacitive) at gate/electrolyte interface upon TGF- β 1 binding, but not the altered charge transfer kinetics from the MB redox reporter" (Line 239-240). This statement should be supported by experimental results (e.g., impedances of the gate electrode before and after TGF- β 1 binding).

Response: We thank the reviewer for the comment. This comment is similar to comment 1 of reviewer 1. Please also refer that answer. We performed the EIS measurement of the Au electrode modified by aptamer with MB redox reporter, and compare the impedance change of

the electrode with different concentration of TGF- β_1 as shown in **Figure R1**. We compare the Impedance change at frequency equal to 1Hz with two different DC offset voltage (0V and -0.3V). Since the redox potential of MB is around -0.2V~-0.3V vs Ag/AgCl, so the change of impedance when the DC offset is -0.3V is mainly attribute to the altered charge transfer resistance between MB and electrode surface (caused by the binding induced conformational change). While the impedance changes when the DC offset is 0V is mostly attributed to the modulation of interfacial capacitance change after target binding. For the original results from the manuscript, the V_G has not been scanned passed the redox potential of MB in the 2E-OECT configuration, so it's reasonable to assume the change of channel current modulation in the 2E-OECT-based E-AB sensor in the original voltage range is likely due to the increased impedance (capacitive) at gate/electrolyte interface.

Revision made: Based on the comment from the reviewer and the results from new experiment, we have deleted this statement.

3. The selectivity of the 3E-OECT-based E-AB sensor is not well characterized. Can redox active metabolites (e.g., dopamine, uric acid, and ascorbic acid) cause signal response?

Response: We thank the reviewer for the comment. This comment is similar to comment 7 of reviewer 1. Please also refer also to the response above. Usually for a biosensor that is used to sense the protein/cytokine, their selectivity is evaluated with other protein/cytokine interference. In this regard, our sensor shows good selectivity over IFN- γ and TGF- β_3 as shown in **Figure R6**. However, testing against redox active metabolites is also important for a redox-based sensor. In terms of the redox-active metabolite interference, we sequentially challenge the 3E-OECT based E-AB sensor with 100 μ M UA, DA and AA as shown in **Figure R10**. For the results in working electrode, UA, DA and AA didn't cause significant change in the redox peak of MB, while the introduction of DA cause higher current in positive voltage range (\sim 0.2V), which is likely due to the redox reaction of DA itself (at around 0.15~0.2V). In terms of both channel current and peak transconductance, UA, DA, and AA cause a noticeable change, with AA showing the smallest competing effect. The selectivity against those redox active metabolites for our 3E-OECT-based E-AB sensor is not optimal, but we believe it can be further improved with more optimization. For example, we can narrow down the scan range of voltage to more negative region to avoid the redox reaction of those metabolites. We can also use charged polymeric coating on the working electrode to impede the diffusion of those charged molecule (UA, DA and AA) to the electrode surface and improve the selectivity.⁵ It is also possible that the redox active soluble interferents react at the channel rather than the working electrode, and thus affect the MB signal transduction. Further investigation on competition with redox active molecules is indeed interesting and important, and should be explored more deeply in future work.

Figure R10. (a) CV results of the aptamer modified Au working electrode in bare 1xPBS buffer solution and the PBS buffer solution with either 100 μM UA, 100 μM UA+DA or 100 μM UA+DA+AA. (b, c) Channel current and slope of channel current of 3E-OECT based E-AB sensor in bare 1xPBS buffer solution and the PBS buffer solution with either 100 μM UA, 100 μM UA+DA or 100 μM UA+DA+AA.

4. In Figure 4 c,f, the channel current responses of the sensors were extracted by measuring the transfer curves. Can the sensors be used for real-time monitoring (e.g. IDS-time curve with fixed VDS and $V_{g,int}$ upon adding TGF- β 1 with a series of concentration)? And how about the response speed of the sensor?

Response: We thank the reviewer for the comment. Since the modulation of the electron transfer kinetics between MB and the electrode surface upon binding is the mechanism for the sensing and this process is relatively quick. The sensor can be used for real time sensing, while for the testing method, instead of fixing the V_{DS} and $V_{g,int}$ and measuring the I_{DS} -t curve, the $V_{g,int}$ should be scanned continuously and cyclically to pass the redox peak of MB to access this redox information. Either channel current modulation or the peak slope of the channel current during each scan can be plotted as a function of time. In terms of response speed, the bottleneck is the binding process between aptamer and the target (around 30 minutes), not the OECT itself (response time usually in ms scale with the size of PEDOT:PSS we use). To improve response speed, voltage pulses could be applied on the gate electrode⁶ or an alternating current electrothermal flow (ACET) can be integrated with OECT,⁷ and potentially accelerate binding between the aptamer and target.

5. Will the area of the sensing gate electrode effect the sensitivity of the 3E-OECT-based sensors?

Response: Yes. A sensing gate with larger area can have more attached aptamer (more MB), which will give a larger faradic current that can induce higher modulation of the channel current. With higher channel current modulation, the sensitivity will be enhanced. While if the sensing gate area is too large, the capacitive current may be already large enough to fully modulate the channel current, which diminishes the impact from the faradic current from MB redox reporter and may deteriorate the sensitivity. This optimization is important and interesting for future investigation, however, it is important to note that a small gate electrode size enables miniaturization for in vivo implementation, and amplifying signals with OECTs becomes increasingly important with small sensing electrodes.

6. It is notably that the absolute current change of the 3E-OECT-based E-AB sensors is much higher than that of the E-AB sensor, while they show a similar detection limit. Are there any strategies to further improve the detection limit of the 3E-OECT-based E-AB sensors?

Response: We thank the reviewer for the comment. This comment is similar to comment 6 of reviewer 1. Please also refer that response. In our design, the sensing is happened in the working electrode that is decoupled from the amplification in the OECT channel, it is well documented on how to improve the detection limit by modifying the working electrode; for example, changing the plain Au to nanostructured Au.⁸ Because increasing the surface area of working electrode is generally accepted as a method to improve the detection limit of an electrochemical biosensor, we believe it is also adaptable to our 3E-OECT-based sensor to improve detection limit.

7. Comparison of the proposed TGF- β 1 sensor with other techniques used for TGF- β 1 sensing is suggested to be provided to show the advantages of the device.

Response: We thank the reviewer for the comment, and we have added this comparison table (Table R1) into our manuscript. Overall, our 3E-OECT based E-AB sensor shows much higher sensitivity compared with all other technique, which further demonstrate the amplification property of OECT. While for detection limit, indeed we show higher detection limit, while we think there is room to improve. In terms of comparison with sensors in each individual paper, for reference 9 and 10, the device they use is exactly same as the electrode-based E-AB sensor we have shown in our manuscript. As expected, our sensor shows much higher sensitivity and similar detection limit. For reference 11, they also use similar approach as electrode-based E-AB sensor while with a graphene/AuNPs modification of the Au working electrode. This nanomaterial modification enables them to have lower detection limit than us; however, our sensor still outperforms in sensitivity. Their finding is also relevant to comment 6 from the reviewer, and it's also possible to improve the detection limit of our sensor by using nanomaterials modified working electrode (gate). For reference 12~15, although they show good detection limit, sophisticated sandwiched modification assay is used, and the in-vivo implementation may present additional challenges. Specifically, sensors in reference 12 and 14 use dissolved redox probes in solution, while sensors in reference 13 and 15 use H₂O₂ to react with the material on the sensing electrode. Both dissolved redox probe and H₂O₂ limited their applicability for in-vivo implementation, which is very important as a future direction.

Revision made: Based on the comment from the reviewer, we have added Supplementary Note 2 and Supplementary Table 1 in Supplementary Information.

sensing technique	electrode modification	recognition element	sensitivity	detection limit	potential in-vivo application	ref.
SWV	bare Au	aptamer	~42 nA/dec	~1 ng/mL	Y	9
SWV	bare Au	aptamer	~42 nA/dec	~1 ng/mL	Y	10
SWV	graphene+ AuNPs	aptamer	~2 μ A/dec	~0.025 ng/mL	Y	11
SWV	bare Au	peptide	143.3 μ A/dec	0.011 ng/mL	N	12
CA	SWCNT	antibody	332 nA/dec	0.95 pg/mL	N	13
EIS	bare Au	antibody	NA	0.57 ng/mL	N	14
CA	SWCNT	antibody	~300 nA/dec	1.3 pg/mL	N	15
CV/SWV-OECT	bare Au	aptamer	292/290 μ A/dec	1 ng/mL	Y	this work

Table R1. Summary of the performance of different biosensors for TGF- β 1 sensing

8. There are many organic mixed ionic–electronic conductors used for OECTs. Why PEDOT:PSS was chosen as the channel materials (counter electrode) in this 3E-OECT-based sensor? Are there any specific considerations in choosing channel materials when design the 3E-OECT-based sensor.

Response: Response: We thank the reviewer for the comment. We choose PEDOT:PSS because it is a stable and commercially available material. In addition, OECTs with PEDOT:PSS channel provide large channel currents, which can lead to higher sensitivity. Last but not least, since the redox potential of MB is around $-0.2\text{V}\sim-0.3\text{V}$ vs Ag/AgCl, we require a material that shows sufficient current modulation in that range, which PEDOT:PSS is capable of. That said, it is likely that significant performance gains can be made by exploring different active materials for the channel. For the considerations in choosing channel materials, the most important criterion is to match the operation region of this material with the redox potential of the redox reporter. For example, p-type accumulation mode materials with threshold voltage around 0V with MB redox reporter; n-type accumulation mode materials with threshold voltage around 0V with ferrocene redox reporter.

Revision made: Based on the comment from the reviewer, we have added this content in page 11 as: “In addition, channel material design rules of the 3E-OECT-based E-AB sensor are such that the operation region of the OECT must be matched with the redox potential of the redox reporter, among other considerations. For example, accumulation mode OECT with p-type channel can work with a MB redox reporter while accumulation mode OECTs with n-type channels are compatible with ferrocene redox reporter.”

Minor:

9. Please specify the V_{DS} in Figure 2c.

Response: We thank the reviewer for the comment and have added the V_{DS} for Figure 2c.

Revision made: Based on the comment from the reviewer, we have added the V_{DS} value (-0.2V) in Fig.2.

10. The $V_{g,int}$ used for extracting the channel current response shown in Figure 4 c, f should be specified, as the channel current response depends on $V_{g,int}$.

Response: We thank the reviewer for the comment and have specified the $V_{g,int}$ for extracting the channel current modulation in Figure 4c and f.

Revision made: Based on the comment from the reviewer, we have added this sentence “The ΔI_{DS} of 3E-OECT-based E-AB sensor is extracted at $V_{g,int}$ equal to -0.4V .” in the caption of Figure 4.

Reviewer #3:

The manuscript reports on an electrochemical aptamer sensor that exploits OECT based amplification to increase the sensitivity. The device enables superior detection of biomedically relevant marker proteins making the approach very interesting for low-cost point of care sensors. The manuscript gains in novelty as the proposed OECT sensor architecture is reported for the first time and can be generally applied to introduce OECT amplification in

amperometric electrochemical sensors. Such a topic is very timely with strong interest from many different research groups. As the contribution is original and well documented I fully support publication of the manuscript in Nat. Commun. However, I have some critical remarks that require a major revision mainly aiming at increasing the clarity of the manuscript and to further support its major claims:

We thank the reviewer for the positive feedback. We performed additional experiments and hope the reviewer's concerns are addressed by the supplementary information.

1. The authors have to address the working mechanism of their OECT enhanced sensor with more detail and clarity. In the current form, the arguments are too simplistic and make it difficult for specialists to understand the work upon first reading. For example in Line 137 the authors argue that "Since the current at the working electrode is equal to the current in counter electrode, which induces the ion injection into the counter electrode that dictates conductivity, we can directly relate the output of the OECT to the current in working electrode in the E-AB sensor." Although in very general terms the argument is correct, it obscures the main working principle enabling the amplification. In particular, a continuous and constant WE current would cause a continuous change in OECT channel current as explained later. The authors should at least clarify here in the initial part of the manuscript that the OECT counter electrode acts as a capacitively coupled electrode and make reference to the later derivation of the underlying eqn. For readers with a background in electronic engineering it would be helpful to make the reference to a charge sensitive amplifier....

Response: We thank the reviewer for the very helpful suggestion. We have indicated that the OECT counter is used as a capacitively coupled electrode. We also changed text to make the working mechanism of our device more clear.

Revision made: Based on the comment from the reviewer, we revised the sensing mechanism in page 4 to 5 as: "Considering the circuit from modified Au working electrode to PEDOT:PSS counter electrode, the current and total charge (time integral of current) passing through the working electrode is equal to that in counter electrode. In this regard, the PEDOT:PSS counter electrode serves as a capacitively coupled electrode whose conductivity can be modulated by the doping/de-doping of the channel active materials resulting from ion injection/extraction, which is driven by the current from working electrode. Since the change of the conductivity in the PEDOT:PSS counter electrode leads to variation of channel current in the OECT, we can directly relate the output of the OECT to the current in working electrode in the E-AB sensor (also considered as gate current I_G from the perspective of the OECT)."

2. Considering the proposed operation mechanism, it is also very surprising that the authors distinguish between source and counter electrode. Is there any reason for this? I expect that the lithographic pattern separating source and CE is not necessary. Such a simplification would enormously enhance understanding of the device.

Response: We thank the reviewer for the comment. This comment is similar to comment 2 of reviewer 1. Please also refer that response. We use two instruments to measure our device such that we can directly compare different modes of operation using the same patterned components: potentiostat for CV/SWV and sourcemeter for OECT. Since the majority of the data in the manuscript was collected with the separated S and CE configuration, we opt to keep this in the main text. However, we make additional note in the text and supporting information

to demonstrate the simplification with combined S and CE yields nearly identical results which would indeed simplify implementation.

3. The OECT-counter electrode approach should be demonstrated with a model experiment in which the potentiostat is operated in galvanostatic mode. Controlled current pulses subjected into the working electrode should then be related to IDS current increase. Overall, the predicted linear relation between injected Q and measured Δ_{IDS} (eqn.1) should be experimentally demonstrated in this way.

Response: We thank the reviewer for the suggestion. We have conducted additional experiment by operating OECT in a galvanostatic mode. As shown in Figure R10, the channel current modulation scale linearly with injected charge (Q) in the range of operation.

Revision made: Based on the comment from the reviewer, we have added the sentence in page 10 as: “Operation of the OECT in galvanostatic mode with constant gate current further support our claim that the modulation of channel current scales linearly with the amount of injected charge (Q) in the region of operation of interest (Supplementary Fig. 17).” We also have added Supplementary Fig. 17 in Supplementary Information.

Figure R11. (a) OECT channel current change with different constant gate current (± 1 nA, ± 2 nA, ± 3 nA, ± 4 nA, ± 5 nA) for 10s each. (b) OECT channel current change with ± 1 nA gate current for different time (5s, 10s, 20s, 30s).

4. A crucial parameter for CV experiments is the scan rate. Scan rates used in the described experiments are not reported. How does the scan rate impact on the observed amplification provided by the 3E-OECT device? This point should be discussed and related experimental data should be shown (at least in Suppl. Mat.).

Response: We thank the reviewer for the comment. We indicated the scan rate in Figure 2 as 60 mV/s. As for the influence of scan rate on the operation of 3E-OECT device, we have performed additional experiments, and the results are shown in **Figure R12**. As expected, for the CV results in electrode, both the amplitude and the separation of reduction and oxidation peaks for MB is enlarged with faster scan rate. For the OECT, the original channel current is recorded as a function of time (only the channel current during reduction scan of MB redox reporter is shown here) and when the channel current is plotted vs. time, quicker change of channel current as well as higher peak slope can be observed with faster scan speed, as expected. When converting the time to voltage (by dividing the scan speed), and plotting the channel current vs. $V_{g,int}$, scan rate of CV doesn't influence the channel current or the slope of channel current too much. The peak position of the slope of channel current does shows similar negative shift as that for the reduction peak of electrode.

Revision made: Based on the comment from the reviewer, we have added the sentences in page 7 as: “We also evaluate how the scan speed, a crucial parameter for CV experiment, influence the behavior of 3E-OECT-based E-AB sensor as shown in Supplementary Fig. 10. As expected, for the CV results in electrode-based sensor, both the amplitude and the separation of oxidation and reduction peaks for MB increase with faster scan rate. For the OECT, the original channel current is recorded as a function of time (only the channel current during reduction scan of MB redox reporter is shown here.) and when the channel current is plotted vs. time, quicker change of channel current as well as higher peak slope can be observed with faster scan speed as expected. When converting the time to voltage by dividing the scan speed and plotting the channel current vs. $V_{g,int}$, scan rate of CV doesn't influence the channel current or the slope of channel current too much. The peak position of the slope of channel current does shows similar negative shift as that for the reduction peak of electrode.” We also have added Supplementary Fig. 10 in Supplementary Information.

Figure R12. (a) Influence of scan rate of CV on the reduction and oxidation peaks of MB redox reporter in an aptamer modified Au working electrode. (b, c) Channel current and slope of channel current for 3E-OECT with different scan rate when plotted as a function of time. (d, e) Channel current and slope of channel current for 3E-OECT with different scan rate when plotted with respect to $V_{g,int}$. Only the channel current during reduction scan of MB redox reporter is shown here.

5. Overall, the comparison with the 2E-OECT is interesting and relevant, given that this configuration is often found in literature. However, I would give priority to a thorough introduction of the new device concept and only later compare 3E-OECT with 2E-OECT as done in figure 3.

Response: We thank the reviewer for the suggestion, and we have reorganized the main text to emphasize the new device concept of 3E-OECT based E-AB sensor.

Revision made: Based on the comment from the reviewer, we have moved the section related to the comparison between 3E and 2E-OECT-based E-AB sensor to the location in the text after the discussion of the operation of 3E-OECT-based E-AB sensor.

6. In the discussion section, a quantitative treatment of the amplification is derived. This should be compared with other articles that quantify amplification of OECT based electrochemical sensors, such as Braendlein, M. et al. Adv. Sci. 2017 or Bonafe et al. Nat. Commun. 2022 et al.

Response: We thank the reviewer for the comment. Quantifying how the OECT amplifies signal in different application is definitely an interesting topic. They and we are trying to study the amplification of OECT in different aspects and implementation. Braendlein et al. explore how the OECT amplifies voltage signals in a voltage amplifier circuit with load resistance, while Bonafe et al. studies how the OECT amplify AC current signal from an electrode. In the work from Bonafe et al., they only consider the AC transport regime where the gate current is capacitive current without Faradaic contributions. For our work, we try to find how the OECT amplifies the time integral of gate current, including both the contribution from capacitive and faradic current. The aim of all these works is to enable us to have a better understanding about how the OECT function as an amplifier. In the revised manuscript, we have added additional discussion about the comparison between our equation and the equation in the references that the reviewer has mentioned.

Revision made: Based on the comment from the reviewer, we have added the sentences in discussion section as: “This equation describes the amplification properties of an OECT to the integral of I_G , which complements the OECT amplification work of others. For example, the work by Braendlein et al. reports on the use of OECT to amplify input voltage in a voltage amplifier circuit with load resistance, and Bonafe et al. shows OECT amplification of AC current signal in the capacitive-dominated regime.” with reference cited.

Minor points:

7. Figure 4c: The y-axis label should be without units.

Response: We are sorry the misleading figure. The y axis in Figure 4c does not describe the ratio between ΔI_{DS} and $I_{pp,CV}$, but the real value of either ΔI_{DS} for 3E-OECT or $I_{pp,CV}$ for working electrode.

Revision made: Based on the comment from the reviewer, we have changed the y-axis in Figure 4c and f from $\Delta I_{DS}/I_{p-p CV}$ and $\Delta I_{DS}/I_{p SWV}$ to $\Delta I_{DS}, I_{p-p CV}$ and $\Delta I_{DS}, I_{p SWV}$

8. The variation in OECT drain current is denoted by I_{DS_m} . This is unusual. The author's should use the delta symbol Δ throughout the manuscript to make clear that here a change in current is described.

Response: We thank the reviewer for the comment and have replaced the I_{DS_m} by ΔI_{DS} .

Revision made: Based on the comment from the reviewer, we have changed I_{DS_m} to ΔI_{DS} .

9. The introduction makes reference to “the typical sensing mechanism of E-AB sensors”. But such a typical mechanism is not described. Is it the variation of an electron transfer rate to a redox moiety in the Aptamer? A better definition should be added to the introduction.

Response: We thank the reviewer for the comment, the sensing mechanism of E-AB can be found on page 2 line 44-46 in the original manuscript. Reproduced below: “Electrochemical detection offers sensitive readout of binding-induced conformation changes, often transduced via changes in electron transfer between an electrode and a redox reporter bonded to the aptamer.”

- 1 Arroyo-Currás, N. *et al.* Real-time measurement of small molecules directly in awake, ambulatory animals. *Proceedings of the National Academy of Sciences* **114**, 645-650 (2017).
- 2 Arroyo-Currás, N. y. *et al.* Subsecond-resolved molecular measurements in the living body using chronoamperometrically interrogated aptamer-based sensors. *ACS sensors* **3**, 360-366 (2018).
- 3 Pong, B. K., Trout, B. L. & Lee, J. Y. Preparation of DNA-functionalised CdSe/ZnS quantum dots. (2007).
- 4 Liu, Y., Tuleouva, N., Ramanculov, E. & Revzin, A. Aptamer-based electrochemical biosensor for interferon gamma detection. *Anal. Chem.* **82**, 8131-8136 (2010).
- 5 Liao, C., Mak, C., Zhang, M., Chan, H. L. & Yan, F. Flexible organic electrochemical transistors for highly selective enzyme biosensors and used for saliva testing. *Adv. Mater.* **27**, 676-681 (2015).
- 6 Liu, H. *et al.* Ultrafast, sensitive, and portable detection of COVID-19 IgG using flexible organic electrochemical transistors. *Science advances* **7**, eabg8387 (2021).
- 7 Koklu, A. *et al.* Convection Driven Ultrarapid Protein Detection via Nanobody-Functionalized Organic Electrochemical Transistors. *Adv. Mater.* **34**, 2202972 (2022).
- 8 Downs, A. M. *et al.* Nanoporous gold for the miniaturization of in vivo electrochemical aptamer-based sensors. *ACS sensors* **6**, 2299-2306 (2021).
- 9 Matharu, Z. *et al.* Detecting transforming growth factor- β release from liver cells using an aptasensor integrated with microfluidics. *Anal. Chem.* **86**, 8865-8872 (2014).
- 10 Zhou, Q. *et al.* Liver injury-on-a-chip: microfluidic co-cultures with integrated biosensors for monitoring liver cell signaling during injury. *Lab on a Chip* **15**, 4467-4478 (2015).
- 11 Gao, Y. *et al.* A flexible multiplexed immunosensor for point-of-care in situ wound monitoring. *Science Advances* **7**, eabg9614 (2021).
- 12 Zhou, L. *et al.* An extracellular matrix biosensing mimetic for evaluating cathepsin as a host target for COVID-19. *Anal. Chim. Acta* **1225**, 340267 (2022).
- 13 Sánchez-Tirado, E. *et al.* Viologen-functionalized single-walled carbon nanotubes as carrier nanotags for electrochemical immunosensing. Application to TGF- β 1 cytokine. *Biosensors and Bioelectronics* **98**, 240-247 (2017).
- 14 Yao, Y. *et al.* Biomarkers of liver fibrosis detecting with electrochemical immunosensor on clinical serum. *Sensors Actuators B: Chem.* **222**, 127-132 (2016).
- 15 Sánchez-Tirado, E., González-Cortés, A., Yáñez-Sedeño, P. & Pingarrón, J. Carbon nanotubes functionalized by click chemistry as scaffolds for the preparation of electrochemical immunosensors. Application to the determination of TGF-beta 1 cytokine. *Analyst* **141**, 5730-5737 (2016).

REVIEWER COMMENTS

Reviewer #1 (Remarks to the Author):

The authors devoted great effort to the revision of the manuscript and it improved considerably. I am very convinced that aptamer based OECT sensors have great potential for advancing current sensor technology, however I didn't fully understand why the authors insist on the term 3E-OECT over 2E-OECT although they demonstrated themselves that the third electrode isn't required and that the main difference between 3E-OECTs and 2E-OECTs is the introduction of an independent reference electrode. I am afraid that this could confuse parts of the readership. Honestly, I also don't fully understand the different signals in Supporting Figure 11 for 3E and 2E. The current that is measured supposed to be given by the number of immobilized redox probes. One possible explanation could be that the impedance of the joined counter/reference electrode in 2E was higher than that of the working electrode since the electrode size of the CE was much smaller than that of the Au working electrode. In this case, current would be limited by the CE, which shouldn't be. Please reevaluate this issue before publication.

Reviewer #2 (Remarks to the Author):

The authors have addressed my most comments. My only concern is that the reasons why "the redox peak current in 2E setup is much lower than that of the 3E setup in both CV and SWV." (Supplementary Fig. 11) remain unclear. If the underlying reasons for the experimentally observed higher sensitivity of the 3E design than that of the 2E-OECT based E-AB sensor can be explained more clearly, the paper can be accepted for publication.

Reviewer #3 (Remarks to the Author):

The authors replied meticulously to all critiques given by the reviewers. Remaining concerns were clarified, and the language was adapted to suit reviewers coming from different fields such as electrochemistry or electronic engineering. Accordingly, I am convinced that the manuscript will become a reference paper on electrochemical sensor development, and I fully support publication in its current form.

Response to reviewers:

We thank the reviewers for their careful consideration and helpful comments. Below we address their questions and comments, introducing new experiments and describing additions or changes in the text.

Reviewer comments are in black.

Our responses in red.

Additions made to the text are in highlighted red.

Reviewer #1 (Remarks to the Author):

The authors devoted great effort to the revision of the manuscript and it improved considerably. I am very convinced that aptamer based OECT sensors have great potential for advancing current sensor technology, however I didn't fully understand why the authors insist on the term 3E-OECT over 2E-OECT although they demonstrated themselves that the third electrode isn't required and that the main difference between 3E-OECTs and 2E-OECTs is the introduction of an independent reference electrode. I am afraid that this could confuse parts of the readership. Honestly, I also don't fully understand the different signals in Supporting Figure 11 for 3E and 2E. The current that is measured supposed to be given by the number of immobilized redox probes. One possible explanation could be that the impedance of the joined counter/reference electrode in 2E was higher than that of the working electrode since the electrode size of the CE was much smaller than that of the Au working electrode. In this case, current would be limited by the CE, which shouldn't be. Please reevaluate this issue before publication.

Response: We thank the reviewer for the positive feedback on our last round of revision and we would like to answer the additional questions that the reviewer has raised. First, in Supplementary Figure 12, we do show that the device has similar characteristics when tested by either merging the S and CE as one electrode (in this case, only two electrodes are contacted to the PEDOT:PSS channel/counter) or separating them as two individual contacts (in this case, three electrodes are contacted with the PEDOT:PSS channel/counter). However, this is not related to the term of 3E-OECT and 2E-OECT. In our original nomenclature, the channel of the OECT is considered as a "terminal" of the electrochemical characterization: akin to one of the electrodes – for example in an OECT, the ionic circuit traverses the electrolyte from gate to channel. As such, the reason we originally named our modified device as "3E-OECT" is because it is tested in a manner which is analogous to the typical 3-electrode setup in electrochemical methods (WE=Gate, RE, CE=Channel). While for conventional implementation of an OECT sensor, it is more akin to a 2-electrode setup in echem, where the channel functions as both RE and CE, and the gate is the WE. After seeing the confusion raised by both reviewers 1 and 2, and through further discussion, we see the point of confusion in spanning the electrochemistry community and the transistor community. As a result, we have elected to implement the suggestion of the reviewer, and to rename these cases: "3E-OECT" to "referenced OECT (ref-OECT)" and "2E-OECT" to "conventional OECT (conv-OECT)". We believe this helps to clarify that our implementation critically includes an independent reference electrode, as compared to the traditional/convention case where there is no reference. We have also clarified some of the language describing the architectures or comparing them in the text.

Second, we appreciate the in-depth critiques from the reviewer regarding the results from Supplementary Figure 11. Indeed, the total charge (time integral of redox current) transferred between the redox reporter and the electrode should be similar in both 3E and 2E testing setup,

and it is determined by the total number of immobilized redox reporters on the electrode like the reviewer suggested. To prove this, we have plotted the anodic scan of the CV (results from Supplementary Figure 11) as a function of time as shown in **Figure R1**. By integrating the current vs. time, we find that the oxidation peak area is 27.4 nC and 22.5 nC from 3E and 2E, respectively. We consider that these values are quite comparable especially given the challenges in differentiating capacitive and faradaic currents and the resulting challenges in baseline assumptions.

In terms of the redox peak height in Supplementary Figure 11, the decreased redox peak height in 2E measurement compared with 3E measurement has also been observed in other literature before.^{1,2} We believe it could be due to the non-linear evolution of potential in working electrode at 2E setup, which is directly related to the height of redox peak. In 3E measurement, the applied voltage is equal to the voltage in working electrode, so the speed of potential evolution in the working electrode is equal to the sweep rate in the entire device (scan speed in CV and scan frequency in SWV). In the 2E measurement, however, the speed of potential evolution at the working electrode is not directly controlled (since there is no independent reference electrode), and the potential is often pinned because of the redox process consuming charge. This process may therefore slow down the change of the potential at the working electrode/electrolyte interface thus broadening and flattening the peak, despite a constant overall applied sweep rate. In addition, considering the fact described above that the total redox charge transferred between redox reporter and working electrode are comparable in 3E and 2E setups, we believe the above explanation helps to support the findings in Supplementary Figure 11.

Revisions made: Based on the comment from the reviewer, we have changed all the “3E-OECT” into “ref-OECT” and “2E-OECT” into “conv-OECT” in the main text.

We have added this sentence in page 8 of the revised manuscript as “...consistent with previous reports.^{38, 39} This phenomenon is likely because the evolution of the potential at the working electrode/electrolyte interface is not rigorously controlled in the 2E setup. This allows potential pinning due to the redox reaction which subsequently broadens and flattens the peak (when plotted against simple applied voltage across the entire device), but the total charge associated with the redox process is roughly equivalent in both 3E and 2E setups (Supplementary Fig. 12).” We also have added the Supplementary Fig. 12 into supplementary information.

Figure R1 (Supp Fig. 12). Time integral of anodic peak of MB in both 3E and 2E testing setup. Oxidation peak area are comparable for 3E (27.4 nC) and 2E (22.5nC) after considering potential contribution from capacitive charging current.

- 1 Graś, M., Kolanowski, Ł., Wojciechowski, J. & Lota, G. Electrochemical supercapacitor with thiourea-based aqueous electrolyte. *Electrochem. Commun.* **97**, 32-36 (2018).
- 2 Hasan, M. R. *et al.* Electrochemical Aptasensor Developed Using Two-Electrode Setup and Three-Electrode Setup: Comparing Their Current Range in Context of Dengue Virus Determination. *Biosensors* **13**, 1 (2022).

Reviewer #2 (Remarks to the Author):

The authors have addressed my most comments. My only concern is that the reasons why “the redox peak current in 2E setup is much lower than that of the 3E setup in both CV and SWV.” (Supplementary Fig. 11) remain unclear. If the underlying reasons for the experimentally observed higher sensitivity of the 3E design than that of the 2E-OECT based E-AB sensor can be explained more clearly, the paper can be accepted for publication.

Response: We thank the reviewer for the comment and appreciate the positive feedback. This comment is similar to the comment from reviewer 1, which we have addressed above. We hope this helps to clarify the findings in Supplementary Fig. 11, and thus helps to explain the higher observed sensitivity.

Reviewer #3 (Remarks to the Author):

The authors replied meticulously to all critiques given by the reviewers. Remaining concerns were clarified, and the language was adapted to suit reviewers coming from different fields

such as electrochemistry or electronic engineering. Accordingly, I am convinced that the manuscript will become a reference paper on electrochemical sensor development, and I fully support publication in its current form.

Response: We thank the reviewer for taking time to review our work and for the positive comments and affirmation to our work.

REVIEWERS' COMMENTS

Reviewer #1 (Remarks to the Author):

The authors have addressed all concerns and the revised manuscript can be considered for publication.

Reviewer #2 (Remarks to the Author):

The revised version is OK for publication.

Reviewer #1 (Remarks to the Author):

The authors have addressed all concerns and the revised manuscript can be considered for publication.

Response: We thank the reviewer for carefully reviewing our manuscript and for the positive assessment of our work.

Reviewer #2 (Remarks to the Author):

The revised version is OK for publication.

Response: We thank the reviewer for reviewing our manuscript and appreciate the positive feedback from the reviewer.